# Examination of the accuracy of SHRIMP U-Pb geochronology based on samples dated by both SHRIMP and CA-TIMS

Charles W. Magee, Jr.[1], Simon Bodorkos[1], Christopher J. Lewis[1], James L. Crowley[2], Corey J. Wall[3], Richard M. Friedman[3]

[1]Geoscience Australia, Cnr Jerrabomberra Ave & Hindmarsh Drive, Symonston ACT 2609 Australia
[2]Department of Geosciences, Boise State University, Boise, Idaho 83725, USA
[3]Pacific Centre for Isotopic and Geochemical Research, Department of Earth and Ocean Sciences, University of British Columbia, 6339 Stores Road, Vancouver, British Columbia V6T 1Z4, Canada

*Correspondence to*: Charles W. Magee, Jr.  (charles.magee@ga.gov.au)

**Abstract.** Estimations of the reproducibility of U-Pb ages from SHRIMP instruments are based on data from studies nearly two decades old. Since that time, refinement of analytical procedures and operational improvements have reduced the historically identified uncertainties of SHRIMP U-Pb analysis. This paper investigates 36 SHRIMP-TIMS double-dated "real-world" geologic samples from a variety of igneous rock types to better understand both geological and analytical sources of disagreement between the two dating methods.

Geoscience Australia's (GA) use of high-precision chemical abrasion thermal ionization mass spectrometry (CA-TIMS) for chronostratigraphy in Australian sedimentary basins has produced a substantial selection of precisely dated zircons, which we can use to cross-correlate the SHRIMP and CA-TIMS ages throughout the Phanerozoic. Thirty-three of the 36 ages were reported with external SHRIMP uncertainties less than 1% (95% confidence). Six of eight cases where the CA-TIMS age was outside the SHRIMP uncertainty envelope were in samples where the 95% confidence interval of the reported SHRIMP age was below 0.66% uncertainty, suggesting that SHRIMP analyses of untreated zircon with smaller uncertainties are probably overoptimistic.

The mean age-offset between SHRIMP and TIMS ages is 0.095%, but the distribution appears bimodal. Geological explanations for age discrepancies between SHRIMP and CA-TIMS are suggested by considering intrusive and extrusive age results separately. All but one sample where the SHRIMP age is more than 0.25% older are volcanic. This offset could be explained by the better single-grain age-resolution of TIMS, allowing identification and exclusion of antecrysts from the eruptive population, while SHRIMP does not have a sufficient single-grain precision to deconvolve these populations – leading to an apparent older SHRIMP age. In contrast, SHRIMP ages from plutonic rocks-particularly plutonic rocks from the early Paleozoic- are typically younger than the CA-TIMS ages from the same samples, most likely reflecting Pb loss from non-chemically-abraded SHRIMP zircons, while chemical abrasion of zircons prior to TIMS analysis destroyed or corrected these areas of Pb loss.

# 1 Introduction

## 1.1 Background and previous work

In recent decades, U-Pb geochronology has progressed from being one of many multi-grain geochronological techniques to being the most widespread and trusted method of determining deep time.

As U has two long-lived isotopes, $^{235}U$, and $^{238}U$, there are three potential isotope ratios which can be used for radiometric dating. The $^{206}Pb/^{238}U$ and $^{207}Pb/^{235}U$ daughter/parent ratios directly measure the decay of the respective isotopes of U, and the near-constant ratio of U isotopes allows the $^{207}Pb/^{206}Pb$ ratio also to be used for age determination. This paper deals exclusively with the $^{206}Pb/^{238}U$ system, which is the primary U-Pb method for the Phanerozoic.

One of the most important developments over the last 15 years has been the use of chemical abrasion on single zircons prior to isotope dilution thermal ionization mass spectrometry (CA-TIMS). CA-TIMS has yielded previously unattainable precision and accuracy in the determination of geologic time from the radioactive decay of $^{238}U$ to $^{206}Pb$ (Mundil et al. 2004, Mattinson 2005).

Since this CA-TIMS revolution, there have been relatively few recent studies of the accuracy and precision limits in Secondary Ion Mass Spectrometry (SIMS) geochronology. There are two large radius SIMS instruments which do the bulk of SIMS U-Th-Pb geochronology: The 1270/1280/1300 series (from CAMECA) and the SHRIMP I/II/RG/V series (from the Australian National University (ANU) and Australian Scientific Instruments (ASI)). In recent years, the only new work relating to the accuracy and precision of U-Pb geochronology is Jeon and Whitehouse (2014) for the 1280 instrument. There have been no recent corresponding studies for the SHRIMP. Schmitt and Vazquez (2017), and Schaltegger et al. (2015), put the 2σ reproducibility of SIMS U-Pb geochronology in the 1-2% to 1-3% range, respectively. In both cases, this number seems to be derived from a daisy chain of references ultimately referring to Claoué-Long et al. (1995) and/or Stern and Amelin (2003). Stern and Amelin (2003) state in the abstract that "for both glass and zircon standards, there remained on average about ± 1% (1σ) unaccounted variation per $^{206}Pb^{+}/^{238}U[O_x]^{+}$ analysis." Black and Jagodzinski (2003) state that improving on this will require researchers "to delve as deeply and objectively as possible into the various sources of uncertainty in SHRIMP dating, so that they can be identified, understood and ultimately either minimised or removed altogether."

There are main areas where these various sources of uncertainty have been investigated are the calibration equation which corrects for variable detection efficiency of U and Pb, and the ability of the SIMS instrument to sample isotopically undisturbed volumes of both reference and unknown zircons. At the time Black and Jagodzinski (2003) wrote this, the SHRIMP community knew that isotopic disturbance in reference zircons was a problem. Black et al. (2003) and Black et al. (2004) introduced a variety of new improved reference materials to address this issue.

The calibration equation is considered a source of uncertainty because the reproducibility of $^{206}Pb/^{238}U$ ages in SIMS analyses generally has a much higher excess error component than the $^{207}Pb/^{206}Pb$ ages, which measure two isotopes of the same element. Jeon and Whitehouse (2014) showed that for the 1280 model of large SIMS instrument, the calibration with the least variance correlates the $Pb^{+}/UO^{+}$ ratio to the $UO_2^{+}/UO^{+}$ ratio. However, Jeon and Whitehouse (2014) do not address the issue of accuracy. Calibrations are instrument specific, and no equivalent study to Jeon and Whitehouse (2014) has been done this century for SHRIMP; Claoué-Long et al. (1995) is still the most recent calibration study for SHRIMP, so the $ln(Pb^{+}/UO^{+})$ vs $ln(UO^{+}/U^{+})$ calibration was used both for published SHRIMP data by the authors listed in Table 1, and for the new data we present.

The uncertainty components of the SHRIMP calibration are classified by Stern and Amelin (2003) into three categories of "error". The first, or within spot, errors are simply the uncertainty associated with each analytical spot, and is generally associated with counting statistics, instrumental instability, and/ or sample heterogeneity on the scale of the spot (if the latter two are present). The second class of errors are the within session, or "internal errors" (Stern and Amelin 2003). This is the additional uncertainty required to make the calibration equation statistically meaningful for the (nominally) uniform primary reference material.  It is called the "Spot to Spot Error" in SHRIMP data

reduction software such as SQUID 2 (Ludwig 2009). The third class of errors in Stern and Amelin (2003) are external errors, or errors related to comparing the results of a session to wider geochronology results. These include the standard error on the calibration constant, the uncertainty in the reference age of the zircon, and also the decay constant errors, if the U-Pb SIMS age is compared to ages from systems other than U-Pb. As this paper only discusses the $^{206}$Pb-$^{238}$U isotopic system, we will not consider decay constant errors further. However, the standard error on the calibration constant (1σ er of mean in Table 2) describes how precisely the reference value for the reference zircon

has been reproduced by the calibration. Like all standard errors, this should be inversely proportional to the number of times the standard zircon is measured in any one SHRIMP session,

In order for geochronology to preserve a crystallographic formation age, both the daughter and parent isotopes must remain locked in the crystal lattice. The Temora 2 reference zircon (Black et al. 2004) seems to behave better than previous reference zircons in this regard (figure 1). However, in some zircons, Pb loss from zircon due to crystallographic damage is a major confounding factor in high-precision zircon

dating. Chemical abrasion preferentially removes damaged zircon that may have suffered Pb loss; Mundil et al. (2004) show that the CA-TIMS procedure can increase the apparent age of some zircons by approximately 4%, due to the removal of the areas which have suffered Pb loss. Several attempts at analyzing zircons before and after chemical abrasion using SIMS show similar changes in age from chemical abrasion, which range from 4% to undetectable (Kryza et al. 2012, Kryza et al. 2014, Watts et al. 2016).

How often rectification of Pb loss by chemical abrasion exceeds the precision of the SIMS analysis in unknowns is then a key issue as to

whether SIMS dating of untreated zircon yields the CA-TIMS age. This depends on two things; the prevalence of zircons with Pb loss which cannot be corrected or avoided by spot placement, and the magnitude of that Pb loss relative to the precision of the SIMS analytical method. The CA-TIMS method works by dissolving all of the zircon where crystallographic lattice damage is sufficient to allow Pb loss prior to analysis (Mundil et al. 2004, Mattinson et al. 2005, Huyskens et al. 2016, Widmann et al. 2019). When these areas of damage are not removed, such as in untreated, natural zircon, the inclusion of these damaged areas in the analytical volume can yield an erroneous (usually

younger) age, if the lattice damage has allowed Pb loss. When analysing zircon, SIMS operators aim to use optical, compositional, and/or cathodoluminescence imagery of zircon to avoid these damaged areas, and instead target undamaged zircon by using the ion beam to extract a small ( <1000 μm$^3$) volume of the best-looking near-surface (~1 μm) material for analysis. So if a zircon has lost Pb from damaged areas sufficient to change the age beyond the precision of the SIMS analytical technique, and if the SIMS analyst cannot identify and avoid these areas, then we would expect the SIMS age to be younger than the CA-TIMS age.

In addition to this isotopic disturbance issue, comparing U-Pb ages from TIMS and SIMS requires that zircon from the same zircon crystallization event is dated by both methods. The small volume of individual SIMS analyses, combined with the 1-2% useful yield of Pb ions (Magee et al. 2014) in SHRIMP, means that multiple spot analyses need to be performed to build up the counting statistics necessary for a precise date. The SIMS sputter crater erodes the conductive gold coat on the SHRIMP mount, and zircon is not a conductor, so old SIMS sputter craters can build up electric charge. This charge can change the ion extraction trajectories from nearby craters in a way that

fractionates U and Pb ions. As a result, SHRIMP spots generally need to be spaced out to a degree (e.g. 10 to 20 μm of gold between rastered

areas) such that the several dozen spots which are to be pooled to calculate a date need to be placed on different zircon grains. If these grains crystallized in different geologic events (such as a volcanic eruption entraining older zircons from previous magmatic pulses), and the analyst is unable to determine this visually, then the calculated date will be a mixture of two geologic ages. This is particularly problematic in the intermediate to silicic volcanic rocks of the East Australian coal basins, which comprise a large proportion of the samples in this study. In

the Phanerozoic, CA-TIMS has the precision to differentiate between sub-million year events of this type, while SIMS does not. As the Metcalfe et al. (2015) and Laurie et al. (2016) papers show with high precision CA-ID-TIMS analysis, the presence of antecrysts in some of their samples may explain their differing interpretations of SHRIMP-TIMS comparison. Thus there is at least one analytical (the uncertainty surrounding the mechanism and reliability of the calibration equation) and two geological (Pb loss and multiple indistinguishable crystallization events) problems which could result in TIMS-SHRIMP discrepancies.

Black and Jagodzinski (2003) claimed that improvements in instrumental performance and reference material quality would be required in order to further understand the limits of SHRIMP precision. Figure 1 shows the spot to spot error for all SHRIMP sessions with published zircon data. This shows that the use of the GA SHRIMP and the adoption of the Temora-2 reference zircon have greatly reduced the spot to spot error in the last decade, compared to the previous twenty years. Thus we feel that these SHRIMP data are ideal for comparison with CA-ID-TIMS to assess the accuracy of these more precise data.

As a result of the Metcalfe et al. (2015) and Laurie et al. (2016) studies, as well as other projects distributed throughout the Palaeozoic and Mesozoic, Geoscience Australia has access to a large amount of accurate, high precision CA-TIMS geochronology data for the Phanerozoic. In addition, our SHRIMP laboratory, now in its fifteenth year of operation, has a database of zircon samples dated by SHRIMP. Cross-referencing samples dated by both techniques allows us to assess the accuracy of the SHRIMP U-Pb ages generated in the Geoscience Australia laboratory, relative to the CA-TIMS method. This study consists of 36 samples from 18 different analytical sessions. Thirty-three

of the 36 ages were reported with external SHRIMP errors less than 1%, so this study will provide an assessment as to whether reporting sub-percent level uncertainty can drive SHRIMP-TIMS age differences outside their respective uncertainties, as has been increasingly reported (Webb et al. in Prep). As the samples in our database were collected for geologic reasons, and not as potential reference zircons, this study allows us to assess the precision and accuracy of SHRIMP U-Pb geochronology "in the wild" from rock types and geologic units which are pertinent to current geologic questions.

**1.2 Sample descriptions**

Geoscience Australia databases were searched for samples measured by both SHRIMP and TIMS. TIMS ages acquired before the adoption of chemical abrasion were not used. Neither were data on reference zircons. SHRIMP data was limited to that taken on the SHRIMP IIe at Geoscience Australia between 2008 and 2019. All SHRIMP data used the Temora-2 zircon as the primary U-Pb reference material, and the Black et al. (2004) U/Pb age of 416.8 Ma as the reference age for that zircon. All 36 samples found were Phanerozoic, and just over half

(19) were Permian, illustrating the importance of Permian coal deposits in Australian stratigraphic research.

All GA sample numbers presented here refer to zircon-rich heavy mineral separates which were concentrated by the Geoscience Australia mineral separation laboratory. In some cases grains selected for TIMS analysis were plucked directly from the SHRIMP mount. In others they were selected from the same zircon-rich heavy mineral separation fraction from which the SHRIMP mount zircons were picked.

The samples dated by both SHRIMP and CA-TIMS in this study are summarized in Table 1, and come from the following previous work:


There are a total of 19 Permian-to-early Triassic samples, roughly half of which are from volcanic units in Eastern Australia coal deposits. Nine samples had Permian to lower Triassic CA-TIMS and SHRIMP dates published in Laurie et al. (2016). A further three samples had Permian CA-TIMS dates published in Metcalfe et al. (2015), but were not included in Laurie et al. (2016). Both of these studies use high precision CA-ID-TIMS to constrain a variety of biostratigraphic problems, including absolute ages for stage boundaries in Australia, correlation between basins, and temporal correlation between the east coast coal basins and marine sequences in Western Australia. Samples GA1978294, GA1978295, and GA1978296 are tuffs in the Lightjack Formation from commercial drill core in the Canning Basin. Samples GA2000865, GA2000869, and GA2122750 are tuffs in the Kaloola member of the Bandanna Formation from two commercial drill holes in the Bowen Basin. Sample GA2122736 is a tuff from the Tinowon Formation in one of the same Bowen Basin holes. There are five samples from four separate commercial drill holes and a road cut in the Sydney Basin. The four drill holes, all from the Hunter region, are: Sample GA2005145, the Nalleen Tuff Member, sample GA2031207, a tuff from the Rowen Formation, GA2031203 from the Awaba Tuff Member of the Newcastle Coal Measures, and GA2031204 from the Nobbys Tuff Member of the Newcastle Coal Measures. Sample GA2005209 is from Garie Formation, sampled from a road cut near Wollongong. We publish the SHRIMP data for all the above samples which were not in Laurie et al. (2016).

Three Permian SHRIMP ages from Cross and Blevin (2010) and Brownlow and Cross (2010) were originally published using data reduced with the SQUID 1 data processing software (Ludwig 2001). These have been reprocessed in Squid 2 (Ludwig 2009). The reprocessed SHRIMP data are presented here. Sample GA1683223 is the Dundee Rhyodacite, a volcanic unit at the top of the Wandsworth Volcanic Group, which was dated by SHRIMP and TIMS to resolve issues with an old Rb/Sr date placing it in the Triassic. The Brownlow and Cross (2010) ID-TIMS age predates the adoption of chemical abrasion at UBC, and is not considered further. Samples GA1954029, Parlour Mountain Leucomonzogranite and GA1954030, the Gwydir River Monzogranite (originally called an adamellite in Cross and Blevin (2010)) both intrude the Wandsworth Volcanics, and thus provide a youngest constraint for that group (Cross and Blevin 2010). CA-ID-TIMS data for all these samples appear in Chapman et al. (2022).

The Wandsworth Volcanic Group was later directly dated, and four dates are presented here. Three geographically widely dispersed samples of the Wandsworth Volcanic Group (GA2120074, GA2120076, and GA2120077) have SHRIMP dates by Chisholm et al. (2014). An additional sample of the Emmaville Volcanics, also part of the Wandsworth Volcanic Group, had a SHRIMP date published by Cross and Blevin (2013). The CA-ID-TIMS data for all Wandsworth Volcanic Group samples are in Chapman et al. (2022). Sample GA3081612, described by Chapman et al. (2022) as 19KP02, the informally-named 'Kings Plains ignimbrite,' is within the Wandsworth Volcanic Group.

There are six Devonian samples in this study. Five are felsic volcanic rocks from the Hill End Trough region of eastern Australia, selected for SHRIMP and CA-TIMS analysis based on their good biostratigraphic control and relevance to the Geologic Time Scale (Bodorkos et al., 2012, 2017; see also Gradstein et al., 2020). In the Cowra Trough, the Bulls Camp Volcanics (sample GA1594761) are of earliest Lochkovian age, as the unit is conformably underlain by a terminal Silurian graptolite assemblage, and overlain by rocks containing middle-Lochkovian conodonts (Pogson and Watkins 1998). In the Hill End Trough, a middle Lochkovian age is indicated for the lower Turondale Formation (sample GA1528019; Jagodzinski and Black 1999), because the upper part of the unit incorporates 'detrital' conodonts of late Lochkovian age in allochthonous limestone blocks (Packham et al. 2001). Overlying the Turondale Formation is the Merrions Formation, with three samples from its type-section (samples GA1528025, GA1528027, and GA1530245; Jagodzinski and Black 1999) each assigned an early Pragian age, based on the presence of Pragian index fossils in overlying sedimentary strata (Packham et al. 2001). Although

Bodorkos et al. (2012, 2017) summarised these CA-TIMS and SHRIMP dates in abstract form, the underlying analytical data are presented here for the first time. Unusually for SHRIMP data, Bodorkos et al. (2012) presented these dates with an uncertainty envelope which included the uncertainty on the Temora-2 reference ID-TIMS age (see uncertainty treatment section below for details).

The sixth Devonian sample is the subvolcanic porphyritic Yithan Rhyolite (GA2097849), which is associated with tin mineralisation in the Central Lachlan Orogen. Bodorkos et al. (2013) published the SHRIMP results; the CA-TIMS data are presented here.

Lewis et al (2015; 2016) undertook geochronological analysis eight compositionally diverse igneous rocks associated with the Cambrian Mount Stavely Volcanic Complex to better understand the chronology of felsic-intermediate magmatism in the Grampians-Stavely Zone. GA2254431 and GA2254432 are samples of the Buckeran Diorite; a medium-coarse grained, equigranular, hornblende-bearing diorite that

intruded the Glenthompson Sandstone. Samples GA2254430 and GA2254436 are from the Bushy Creek Granodiorite; a variably medium to coarse grained, generally equigranular, slightly quartz-phyric to biotite-bearing granodiorite that intrudes the Glenthompson Sandstone. Samples GA2167510, GA2169175, and GA2172148 are all various subvolcanic porphyritic dacites (both molybdenite-bearing and barren) that cross cut the deformed co-magmatic Mount Stavely Volcanic Complex and older Glenthompson Sandstone. Finally GA1486519, the Narrapumelap Road Dacite Member, is a dark pink-grey dacitic to rhyolitic lava within the Towanway Tuff—part of the Mount Stavely

Volcanic Complex. . Lewis et al. (2015, 2016) published SHRIMP ages and CA-TIMS summaries, while the details of the CA-TIMS analyses are presented here.

In addition to these samples, new data using both SHRIMP and CA-TIMS are presented from two unrelated samples.

Sample GA2121671 is a calcilutite from a petroleum exploration hole, Alaric 1 (2890-2940 mRT), in the Northwest Shelf, Western Australia. This interval was originally logged as Triassic age. Upon further micropaleontological analysis, the sampled interval intersects Triassic to

Cretaceous strata, where the top of the core section is interpreted as Cretaceous-aged Forestier Claystone, while the bottom is interpreted as Triassic-aged Mungaroo Formation (Helby et al. 2004). The interval yielded Cretaceous zircon with no signs of sedimentary transport.

Sample GA1977984 is the nominally Ordovician Saddington Tonalite, in Queensland. The Saddington Tonalite is a term used to group fintrusions of the Gray Creek Complex and Judea Formation which range from diorite to tonalite (Henderson et al. 2011). This sample was dated to supersede an unpublished laser date with multiple age populations.

**2 Methods**

**2.1 Analytical methodology**

**2.1.1 SHRIMP analytical methodology**

The 36 samples whose data are presented here were analysed in a total of 18 different SHRIMP analytical sessions, run by eight different operators, between 2008 and 2019. Additional details for published work are given in the references listed in Table 1. Session-specific

information is given in Table 2. In general, the SHRIMP configuration was typically a 1.6-2.5 nA $O_2^-$ beam, as measured by the Primary Beam Monitor (PBM), which measures net sample current (generally 1.6 times the true primary beam in zircon, when positive secondary ions are extracted). The mass filtered primary beam is projected through a 100 μm aperture to form a ~14x20 μm evenly illuminated spot on the zircon. Although pit depths were not directly measured, the pit depth measurements from Magee et al. (2014) from this instrument

running under the same analytical conditions yield a sputter rate for a 100 μm Köhler aperture of 0.34 nm s$^{-1}$ nA$^{-1}$ of O$_2^-$ primary beam, meaning that the sputtering craters were generally 700-1000 nm deep, with occasional depths as little as 500 nm or as much as 1350 nm.

The secondary mass spectrometer was set up with a 110 μm source slit and a 100 μm collector slit, yielding an average mass resolution of about 5000 at the 1% level for Pb isotope peaks. HfO$_2^+$ was resolved from Pb$^+$ to within the limits of detection. The energy window was approximately -40 to +55 eV, relative to the 10 keV secondary acceleration energy.

In most analytical sessions, the magnet cycled through 10 mass stations 6 times. The masses corresponded to the following species: $^{90}$Zr$_2^{16}$O; $^{204}$Pb; Background ($^{204}$Pb+0.05 amu); $^{206}$Pb; $^{207}$Pb; $^{208}$Pb; $^{238}$U; $^{248}$Th$^{16}$O; $^{254}$U$^{16}$O; $^{270}$U$^{16}$O$_2$. Count times were generally 20 seconds for $^{204}$Pb and background, 15 seconds for $^{206}$Pb, 40 seconds for $^{207}$Pb, and 2-5 seconds on all other peaks. Additional mass stations and altered count times were used in session 110088 (Magee et al. 2017). A calibration slope of two was used for all sessions except 90001 (slope 1.52) and 100127 (slope = 1.77). Common Pb was corrected for using either the $^{207}$Pb (assuming concordance) or the $^{204}$Pb isotope, with $^{204}$Pb generally preferred for older zircons, and $^{207}$Pb for younger ones.

## 2.1.2 TIMS methodology

For the TIMS dates from Boise State University, U-Pb dates were obtained by the chemical abrasion isotope dilution thermal ionization mass spectrometry (CA-TIMS) method, modified after Mattinson (2005), from analyses composed of single zircon grains. Zircon picked from mounts or separates provided by Geoscience Australia was placed in a muffle furnace at 900°C for 60 hours in quartz beakers.

Zircon was put into 3 ml Teflon PFA beakers and loaded into 300 μl Teflon PFA microcapsules. Fifteen microcapsules were placed in a large-capacity Parr vessel and the zircon partially dissolved in 120 μl of 29 M HF for 12 hours at 180°C or 190°C. Zircon was returned to 3 ml Teflon PFA beakers, the HF was removed, and zircon was immersed in 3.5 M HNO$_3$, ultrasonically cleaned for an hour, and fluxed on a hotplate at 80°C for an hour. The HNO$_3$ was removed and zircon was rinsed twice in ultrapure H$_2$O before being reloaded into the 300 μl Teflon PFA microcapsules (rinsed and fluxed in 6 M HCl during sonication and washing of the zircon) and spiked with the Boise State University mixed $^{233}$U-$^{235}$U-$^{205}$Pb tracer solution (BSU-1B) or EARTHTIME mixed $^{233}$U-$^{235}$U-$^{205}$Pb tracer solution (ET535). Zircon was dissolved in Parr vessels in 120 μl of 29 M HF with a trace of 3.5 M HNO$_3$ at 220°C for 48 hours, dried to fluorides, and re-dissolved in 6 M HCl at 180°C overnight. Solutions were subsequently dried down and redissolved in 60 μl of 3 M HCl to convert to PbCl$_3^-$, UO$_2$Cl$_3^-$, and UCl$_6^{2-}$ ions. U and Pb were separated from the zircon matrix using an HCl-based anion-exchange chromatographic procedure (Krogh, 1973). Pb was eluted with 200 μl of 6 M HCl and U with 250 μl of MQ-H$_2$O into the same beaker and dried with 2 μl of 0.05 N H$_3$PO$_4$.

Pb and U were loaded on a single outgassed Re filament in 5 μl of a silica-gel/phosphoric acid mixture (Gerstenberger and Haase, 1997), and U and Pb isotopic measurements made on a GV Isoprobe-T multicollector thermal ionization mass spectrometer equipped with an ion-counting Daly detector. Pb isotopes were measured by peak-jumping all isotopes on the Daly detector for 100 to 160 cycles. Mass fractionation was determined using the ET2535 tracer solution that has $^{202}$Pb and $^{205}$Pb, and thus measures fractionation directly. It was either 0.16 ± 0.03%/a.m.u. or 0.18 ± 0.03%/a.m.u. (1σ), for the analytical sessions reported here. Transitory isobaric interferences due to high-molecular weight organics, particularly on $^{204}$Pb and $^{207}$Pb, disappeared within approximately 60 cycles, while ionization efficiency averaged 10$^4$ cps/pg of each Pb isotope. Linearity (to ≥1.4 x 10$^6$ cps) and the associated deadtime correction of the Daly detector were monitored by repeated analyses of NBS982. Uranium was analyzed as UO$_2^+$ ions in static Faraday mode on 10$^{12}$ ohm resistors for 300 cycles, and corrected

for isobaric interference of $^{233}U^{18}O^{16}O$ on $^{235}U^{16}O^{16}O$ with an $^{18}O/^{16}O$ of 0.00206. Ionization efficiency averaged 20 mV/ng of each U isotope. U mass fractionation was corrected using the known $^{233}U/^{235}U$ ratio of the tracer solution.

U-Pb dates and uncertainties were calculated using the algorithms of Schmitz and Schoene (2007), calibration of BSU-1B tracer solution of $^{235}U/^{205}Pb$ of 77.93 and $^{233}U/^{235}U$ of 1.007066, calibration of ET535 tracer solution (Condon et al., 2015) of $^{235}U/^{205}Pb$ = 100.233, $^{233}U/^{235}U$ = 0.99506, and $^{205}Pb/^{204}Pb$ = 11268, U decay constants recommended by Jaffey et al. (1971), and $^{238}U/^{235}U$ of 137.818 (Hiess et al., 2012). $^{206}Pb/^{238}U$ ratios and dates were corrected for initial $^{230}Th$ disequilibrium using $D_{Th/U}$ = 0.20 ± 0.05 (1σ) and the algorithms of Crowley et al. (2007), resulting in an increase in the $^{206}Pb/^{238}U$ dates of ~0.09 Ma. All common Pb in analyses was attributed to laboratory blank and subtracted based on the measured laboratory Pb isotopic composition and associated uncertainty. U blanks are estimated at 0.013 pg.

Weighted mean $^{206}Pb/^{238}U$ dates are calculated from equivalent dates (probability of fit >0.05) using Isoplot 3.0 (Ludwig, 2003). Errors on weighted mean dates are given as ± x / y / z, where x is the internal error based on analytical uncertainties only, including counting statistics, subtraction of tracer solution, and blank and initial common Pb subtraction, y includes the tracer calibration uncertainty propagated in quadrature, and z includes the $^{238}U$ decay constant uncertainty propagated in quadrature. Internal errors should be considered when comparing our dates with $^{206}Pb/^{238}U$ dates from other laboratories that used the same tracer solution or a tracer solution that was cross-calibrated using EARTHTIME gravimetric standards. Errors including the uncertainty in the tracer calibration should be considered when comparing our dates with those derived from other geochronological methods using the U-Pb decay scheme (e.g., laser ablation ICPMS). Errors including uncertainties in the tracer calibration and $^{238}U$ decay constant (Jaffey et al., 1971) should be considered when comparing our dates with those derived from other decay schemes (e.g., $^{40}Ar/^{39}Ar$, $^{187}Re-^{187}Os$). The 2σ uncertainties were converted to 95% confidence intervals after data delivery to Geoscience Australia in the same manner as the UBC TIMS data, whose analytical method is described next.

For the TIMS analyses performed at the University of British Columbia, the methodology was modified from what is described in Scoates & Friedman (2008). Individual zircon crystals were placed in a muffle furnace at 900°C for 60 hours in quartz beakers to anneal minor radiation damage and prepare the crystals for subsequent chemical abrasion (Mattinson, 2005). Zircon crystals were subjected to a modified version of the chemical abrasion method of Mattinson (2005), whereby single crystal fragments were individually abraded in a single step with concentrated HF. Zircon was put into 3 ml Teflon PFA beakers, rinsed in 3.5 M HNO₃ three times before being loaded into 300 μl Teflon PFA microcapsules. Microcapsules were then placed in a large-capacity Parr vessel and the zircon partially dissolved in 100 μl of 29 M HF for 12 hours at 180°C. Zircon was returned to 3 ml Teflon PFA beakers, HF was removed, and zircon was immersed in 6 M HCl, ultrasonically cleaned for 30 minutes, and fluxed on a hotplate at 80°C for an hour. Zircon was dissolved in Parr vessels in 120 μl of 29 M HF with a trace of 3.5 M HNO₃ at 220°C for 48 hours, dried to fluorides, and re-dissolved in 6 M HCl at 180°C overnight. Solutions were subsequently dried down with 2 μl of 0.05 N H₃PO₄ and are ready for mass spectrometry.

Pb and U were loaded on a single outgassed zone-refined Re filament in 2 μl of a silica-gel/phosphoric acid mixture (Gerstenberger and Haase, 1997), and U and Pb isotopic measurements made on a VG Sector 54S thermal ionization mass spectrometer with Sector 54 electronics equipped with an analog Daly detector. Pb isotopes were measured by peak-jumping all isotopes on the Daly detector for 100 cycles and corrected for 0.18 ± 0.05%/a.m.u. (1σ) mass fractionation from repeated measurements of NBS-982. Transitory isobaric interferences due to high-molecular weight organics, particularly on $^{204}Pb$ and $^{207}Pb$, disappeared within approximately 60 cycles and monitored on masses $^{201}Pb$ and $^{203}Pb$. Uranium was analyzed as $UO_2^+$ ions by peak-jumping all isotopes on the Daly detector for 100 cycles and corrected for isobaric interference of $^{233}U^{18}O^{16}O$ on $^{235}U^{16}O^{16}O$ with an $^{18}O/^{16}O$ of 0.00206. U mass fractionation was corrected using the known $^{233}U/^{235}U$ ratio of the tracer solution.

U-Pb dates and uncertainties were calculated using the algorithms of Schmitz and Schoene (2007), calibration of ET535 tracer solution (Condon et al., 2015) of $^{235}U/^{205}Pb = 100.233$, $^{233}U/^{235}U = 0.99506$, and $^{205}Pb/^{204}Pb = 11268$, U decay constants recommended by Jaffey et al. (1971), and $^{238}U/^{235}U$ of 137.818 (Hiess et al., 2012). $^{206}Pb/^{238}U$ ratios and dates were corrected for initial $^{230}Th$ disequilibrium using a Th/U in the magma of 3. All common Pb in analyses was attributed to laboratory blank and subtracted based on the measured laboratory Pb isotopic composition and associated uncertainty. U blanks are estimated at 0.020 pg. Weighted mean $^{206}Pb/^{238}U$ dates are calculated from equivalent dates (probability of fit >0.05) using Isoplot 3.0 (Ludwig, 2003).

To make uncertainty treatments of all TIMS results consistent with those from Laurie et al. (2016), the uncertainty of the weighted mean was multiplied by the square root of the MSWD, if it was greater than 1, and Student's t, which resulted in an increased uncertainty envelope. This is especially pronounced for those samples where there were only three grains in the mean.

## 2.2 Uncertainty treatment

For the TIMS analyses, the reported uncertainty includes the analytical uncertainty as well as the spike uncertainty (as the data reported here originate from two labs). For the SHRIMP analyses, the reported uncertainty includes the analytical uncertainty of the unknowns, the session mean (the standard error of the individual spot calibration constant measurements made on the reference zircon Temora-2), and the uncertainty on the reference age assigned to the Temora-2 (Black et al. 2004) reference zircon. This is calculated by adding twice the $1\sigma$ spike uncertainty ($2 \times 0.13\% = 0.26\%$) in quadrature to the analytical uncertainty (0.08%) reported by Black et al. (2004), yielding a reference zircon value of $416.78 \pm 1.13$ Ma (an uncertainty of 0.27%). As the uncertainty of reference values of reference materials is rarely incorporated into published uncertainty, the 95% confidence envelopes presented in this paper may differ to those in the original sources. As all measurements take place within the U-Pb system, no uncertainties associated with the $^{238}U$ or $^{235}U$ decay constants are propagated. In all cases, the SHRIMP uncertainty is substantially larger than that from CA-TIMS. The additional 0.707% error incorporated into the Laurie et al. (2016) uncertainties is not included here, as that error covered differences between SHRIMP instruments at different institutions, and the use of different reference zircons. Because all SHRIMP data in this paper were produced at Geoscience Australia using Temora-2 as the primary reference material, this error enhancement serves no purpose, and may obscure subtle systematic effects which we hope to observe. Thus the uncertainties shown in this study are smaller than those reported in Laurie et al. (2016) for the same SHRIMP data.

There are four CA-TIMS analyses whose ages were published by both Metcalfe et al. (2015) and Laurie et al. (2016). Although both papers report numbers from the same analytical sessions, the reported numbers differ because of differences in the reporting of uncertainties (2 σ vs 95% confidence determined using Student's t and sqrt MSWD where greater than 1). We use the Laurie et al. (2016) numbers as the uncertainty calculated using Student's t is more robust. For those samples where the TIMS data has not yet been published, the results are listed in the results section, below. The full datasets for both the new TIMS data and the new SHRIMP data are in the electronic data tables (see results section for details).

## 3 Results

### 3.1 CA-TIMS ages

CA-TIMS analyses of 5-12 individual single zircon crystals from the zircon concentrates from each sample yielded groups of 3-12 concordant grains. These were interpreted as the igneous age of the zircons samples. Only four of the 36 samples had younger outliers, consistent with the hypothesis that chemical abrasion removes zircon which has suffered lattice damage sufficient to cause Pb loss. In contrast, 20 of the samples had inherited grains. Two of the eleven intrusive rocks had inherited grains, as did 18 of the 25 volcanic rocks. This is consistent with the observation in modern volcanoes that zircon ages in any given eruption span the range of the volcanic edifice (Claiborne et al. 2010), and that CA-TIMS dating is sufficiently precise that Palaeozoic zircons hundreds of thousands of years older than the igneous population may be resolvable. Full data from Boise State University are presented in electronic Table S2, while UBC data are presented in electronic Table S3. The new TIMS ages (n = 16) are summarized in the text below to make them more easily searchable in the literature.

### 3.1.1 Mesozoic Northwest Shelf

CA-TIMS results from 7 of 9 zircons from a ditch cuttings sample from the North West Shelf Alaric 1 drillhole (GA2121671) yield an age of 139.15 ± 0.09 Ma (Table S2). The c. 139 Ma zircon age is inconsistent with the Triassic (252-201 Ma) age originally assigned to the core based on microfossils low in the section. If the zircons are from an unrecognised ash-fall layer, then the c. 139 Ma age is the age of deposition. If they are detrital, it is a maximum deposition age. The minimum age, as defined by dinocysts (Helby et al. 2004) in overlying cuttings samples (2655–2720 mRT and 2870–2900 mRT)—upper to lower *Batioladinium reticulatum* Zone (c. 141.4–140.2 Ma) and upper *Cassiculosphaeridia delicata* Zone (143.6–142.6 Ma), respectively—is Berriasian.

Although this stratigraphy appears at first glance to be inverted, the age- stratigraphy correlations of Helby et al. (2004) predate the invention of CA-TIMS. So it is possible that their calibration of fossil stratigraphy to radiometric time calibration is based on obsolete isotopic geochronological measurements. More recently, Lena et al (2019) bracket the Jurassic/Cretaceous boundary at between 140.7 and 140.9 Ma using CA-TIMS, almost 5 million years younger than Helby et al. (2004), and putting our 139.15 Ma age comfortably in the early Berriasian.

### 3.1.2 Devonian (Bodorkos et al. (2012)

Bodorkos et al. (2012) presented the comparison of five SHRIMP and TIMS ages in abstract form. The complete CA-TIMS data from that report are given here.

Five zircons from the Bulls Camp Volcanics (GA1594761) yield a group of three concordant ages and two analyses interpreted to have been affected by Pb loss. (Table S3-B) The three concordant grains yield an age of 417.75 ± 0.88 Ma.

Six zircons from the Turondale Formation (GA1528019) yield an age of 415.56 ± 0.51 Ma (Table S3-C).

Five zircons from the Merrions Formation (GA1528025) yield two older concordant ages which are interpreted as inherited and three younger grains with a combined age of 412.73 ± 0.96 Ma (Table S3-D).

Six zircons from the Merrions Formation (GA1528027) yield two older ages, and a group of four concordant analyses with a combined age of 411.71 ± 0.89 Ma. (Table S3-E).

Five zircons from the Merrions Formation (GA1530245) yield a group of four concordant ages and one younger, less precise zircons which is interpreted as having lost Pb (Table S3-F). The four older grains yield a combined age of 413.76 ± 0.76 Ma.

### 3.1.3 Other Devonian

Seven of eight zircons from GA2097849 (Yithan Rhyolite) yielded an age of 414.27 ± 0.26 Ma (Table S2).

### 3.1.4 Ordovician

Six chemically abraded zircons from the Saddington Tonalite (GA1977984) yield a weighted mean $^{206}Pb/^{238}U$ age of 487.07 ± 0.70 Ma (Table S3-A). Henderson et al. (2011) describe the Saddington Tonalite as a suite of diorite to tonalite intrusions of the Bendigonian graptolite-bearing Judea Formation, and the older Gray Creek formation. The Bendigonian corresponds to the middle Floian in the ICS 2020 timescale (Gradstein et al. 2020). The age of 487.07 ± 0.70 Ma is within uncertainty of the Cambrian-Ordovician boundary, so further work on the geochronology, stratigraphy, and contact relationships of this region may be required, as this particular sample is young enough to intrude the Gray Creek formation, but not the Judea Formation.

### 3.1.5 Cambrian Stavely

Sample GA2254430, the Bushy Creek Granodiorite, yielded a CA-TIMS age of 501.55 ± 0.31 Ma from six of seven zircons (Table S2).
Sample GA2254436, the Bushy Creek Granodiorite, yielded a CA-TIMS age of 501.65 ± 0.29 Ma from six of six zircons (Table S2).
Sample GA2254431, the Buckeran Diorite, yielded a CA-TIMS age of 504.83 ± 0.30 Ma from eight of eight zircons (Table S2).
Sample GA2254432, also the Buckeran Diorite, yielded a CA-TIMS age of 505.00 ± 0.36 Ma from seven of seven zircons (Table S2).
Sample GA1486519, the Narrapumelap Road Dacite Member, yielded a CA-TIMS age of 507.21 ±0.39 Ma from seven of eight zircons (Table S2).
Sample GA2167510, the 'Victor Porphyry', yielded a CA-TIMS age of 504.17 ± 0.31 Ma from five of six zircons (Table S2).
Sample GA2169175, also the 'Victor Porphyry', yielded a CA-TIMS age of 503.80 ± 0.29 Ma from six of six zircons (Table S2).
Sample GA2172143, a molybdenite porphyry, yielded a CA-TIMS age of 504.53 ± 0.31 Ma from eight of eight analyses (Table S2).

## 3.2 SHRIMP ages

### 3.2.1 Secondary standards

The OG1 Paleoarchean zircon has been run on most GA mounts as a monitor for $^{207}Pb/^{206}Pb$ isotopic fractionation since the publication of Stern et al. (2009). Although intended as a $^{207}Pb/^{206}Pb$ isotopic reference material, the data can be reduced to yield a $^{206}Pb/^{238}U$ age. Sixteen of the 18 analytical sessions used in this study (all except 80101 and 110088) had OG1 analyses, and the results are shown in figure 2.
Although the weighted mean OG1 SHRIMP age from the 16 sessions of 3439.6 ± 7.0 Ma (MSWD 1.9, PoF=0.02) is about 0.6% lower than the CA-ID TIMS age, it is well within uncertainty of the NON-chemically abraded ID-TIMS age for OG1 of 3440.7 ± 3.2 Ma (Stern et al.

2009). This result has some excess dispersion, which is entirely due to session 90038 being an older outlier (excluding gives $3437.4 \pm 4.8$ Ma; MSWD=0.84; PoF=0.63). The obvious sub-millimeter damage done to zircons which have undergone chemical abrasion (Huyskens et al. 2016) might be interpreted as suggesting that chemical abrasion is a large scale process wherein damaged crystallographic domains are dissolved away and the zircon that survives chemical abrasion is untouched. The size and morphology of the dissolved areas implies that a microbeam technique should be able to avoid the damaged areas and return the same age as chemical abrasion TIMS (Bodorkos et al. 2009). However, our OG1 data suggest that microbeam targeting of the best-looking areas of untreated zircons return the NON-chemically abraded age for untreated zircons instead of the chemically abraded age (with the possible exception of session 90038, which is within error of the CA age, but not the untreated age). This implies that Pb loss on the scale of a 20x15x1 μm SHRIMP spot usually cannot be avoided in chemically unabraded OG1 by careful targeting.

The only Session whose OG1 $^{206}Pb/^{238}U$ age was not within uncertainty of the untreated reference value was session 90038. However, the unknown measured in this session, GA1977984, was only 0.05% older (and well within error) of the TIMS age. So it is hard to tell if this was instrumental instability or unusually adept spot placement.

A couple of other sessions had additional secondary reference zircons. Session 190057 had 20 analyses of zircon 91500, with a $^{206}Pb/^{238}U$ age of $1063.2 \pm 5.6$ Ma, and MSWD of 0.87, and a probability of fit of 0.62. Session 110088 had 49 analyses of R33, with a $^{206}Pb/^{238}U$ age of $419.7 \pm 1.4$ Ma, and MSWD of 1.15, and a probability of fit of 0.22. These results are within their respective reference values.

### 3.2.2. Unknowns

The 36 SHRIMP ages compiled in this paper were obtained in a total of 18 SHRIMP sessions. The details of the samples and sessions are in Table 1 and Table 2. Data from twelve of those sessions have been previously published. Of those, two (80101 and 80104) were originally published using SQUID1 data reduction software (Ludwig 2001). These have been reprocessed using SQUID 2 (Ludwig 2009). The spot by spot results for new data are in Table S4, and the final ages with comparisons are in Table 3. SHRIMP ages are not listed in the text as we feel the TIMS ages are more appropriate as ages of record for these samples. Additional information is available in the Geoscience Australia Geochronology Delivery Service: http://www.ga.gov.au/geochron-sapub-web/geochronology/shrimp/search.htm

### 4 Discussion

### 4.1 Summary of SHRIMP-TIMS comparison

Table 3 shows 36 samples for which U-Pb dates have been determined using both SHRIMP and CA-TIMS. The difference (SHRIMP age minus TIMS age) per sample is given in terms of Ma, percent of the total CA-TIMS age, and number of SHRIMP 95% confidence intervals. The SHRIMP confidence interval is used because the mean SHRIMP 95% confidence interval is almost eight times larger than the mean CA-TIMS 95% confidence interval, and therefore dominates the uncertainty. In eight of the 36 samples, the TIMS age lay beyond the SHRIMP 95% confidence interval. In five of these the SHRIMP age was younger, and in three the SHRIMP age was older than the TIMS age. The average 95% confidence interval for all 36 SHRIMP results was 0.71%, while the average for the 8 samples where the TIMS age lay outside that interval was 0.66%. This suggests that the SHRIMP confidence intervals are optimistic, as the expected number of the 36 samples to lie outside the 95% confidence interval is 36 x 0.05 = 1.8.

Of the three samples where the reported SHRIMP uncertainty envelope was greater than 1%, one sample had a TIMS age outside the SHRIMP uncertainty envelope (33%). Of the six samples with a SHRIMP uncertainty between 1% and 0.75%, none had TIMS ages outside the uncertainty envelope (0%). Of the 24 samples where the SHRIMP uncertainty envelope was between 0.75% and 0.5%, six TIMS ages were outside the SHRIMP uncertainty envelope (25%). Of the three samples where the reported uncertainty envelope was less than 0.5%, one had a TIMS age outside the SHRIMP uncertainty envelope (33%). In summary, six of the eight cases where the TIMS age lay outside the uncertainty envelope of the SHRIMP data had reported SHRIMP uncertainties of less than 0.66%. This suggests that reported SHRIMP uncertainties below two thirds of a percent are increasingly likely to be inaccurate as the 95% confidence envelope contracts. Possible reasons for this, and potential approaches to overcome this barrier are discussed below.

## 4.2 Possible explanations and approaches

### 4.2.1 Primary beam species

GA2122736 and GA2122750 were analysed on the SHRIMP using an $^{18}O_2^-$ primary beam instead of a $^{16}O_2^-$ beam (Magee et al. 2014, Magee et al. 2017). This does not seem to have resulted in any statistically meaningful change in the SHRIMP result accuracy relative to CA-TIMS, compared to the other 34 samples, which were analysed using $^{16}O_2^-$, as both are within the offset range of the samples measured using $^{16}O_2^-$. Thus we discount this as a factor in the observed offsets.

### 4.2.2 Calibration discussion

The 36 samples were run in a total of 18 analytical sessions, with anywhere from one to five of the samples in this comparison analysed in any one session. The eight samples whose ages disagree beyond their uncertainty envelopes were analysed in a total of 7 sessions, with session 100103 having two samples with TIMS ages outside the SHRIMP uncertainty envelope. These two samples, GA2031203, and GA2031207, differed from the TIMS in opposite directions- the SHRIMP age was older for sample GA2031203, while the TIMS age was older for sample GA2031207.

Not every sample analysed in each of the 18 SHRIMP sessions has been submitted for TIMS. As a result, the number of samples run in each session, and therefore the length of each session, is quite variable. This leads to a wide range in the number of reference zircon analyses run, with anywhere between 17 and 79 Temora-2 analyses. As the calibration uncertainty involves the standard error of the reference zircon analyses, we would expect sessions with more analyses to have tighter uncertainties than those with fewer. A graph of the calibration uncertainty vs the square root of the number of analyses is shown in figure 3a, which shows that there is substantial variability beyond that expected from statistics, and that four sessions in particular- 80101, 10104, 100128, and 110011- define a "bad calibration" trend where the observed uncertainty is much larger than for the other 14 sessions, even when accounting for the number of reference zircon analyses. However, none of the samples whose TIMS and SHRIMP ages disagree were analysed in these bad sessions. All of the mismatched ages come from sessions with well-behaved calibrations, and they are spread all along the x axis from few to many reference zircons analysed (figure 3a).

Figure 3b directly compares the SHRIMP-TIMS difference for each unknown for the calibration error. The spots where the ages disagree are shown in red, and none of them have a calibration error greater than 0.2%. The other figure of merit associated with the calibration is the spot-to-spot error. Figure 3c compares the calculated spot to spot error to the SHRIMP-TIMS difference. Note that for spot-to-spot errors

less than 0.75%, a spot to spot of 0.75% was propagated through to the unknowns. This figure shows medium to low spot to spot errors associated with the disagreeable ages, and none above 1%.

If the unknown zircons are as well behaved as the reference zircons, then a plot of unknowns similar to figure 3a should show a similar pattern, This plot, the square root of the number of unknown analyses vs the unknown total uncertainty, is shown in figure 3d. Unlike the plot of the calibrations (figure 3a) the plot of the unknowns (figure 3d) does not show the low n samples on a similar trend line to moderate to high n samples- they all have larger errors than expected. And while most of the disagreeable samples form a trend along the lowest boundary of the confidence interval, there are two substantial outliers.

To better examine the sources of uncertainty, we plot all three components of sample uncertainty for each sample in figure 4. As these are added in quadrature, we have plotted the squares of each component and stacked them; the total uncertainty is the square root of each value. The square of the SHRIMP-TIMS difference is plotted as a line on each sample; green if it is within uncertainty, red if it is not.

This figure shows that the reference value is generally the smallest component, and is obviously the most consistent. The calibration component of the uncertainty is usually the next smallest component for most (but not all) samples, and has a modest amount of variation.

In no case does a large calibration uncertainty stretch the stack into catching an offset. However, for half a the disagreeable samples, a large calibration uncertainty would have pushed the TIMS and SHRIMP ages into agreement, as the calibration errors for those samples were small, and the TIMS and SHRIMP ages were very close to agreement. Finally, the largest (for most samples) and by far the most variable is the uncertainty related to the analysis of the unknowns. This suggests that examining the unknowns may yield the source of disagreement instead of analysing the calibration.

**4.2.3 Outlier discussion**

In only two samples did the SHRIMP and TIMS ages differ by more than 1%. One of these, sample GA2031207, is a Guadalupian tuff from the Rowan Formation, with a TIMS age of $271.60 \pm 0.13$ Ma, where the SHRIMP date is younger, at $268.7 \pm 1.9$ Ma. An earlier analysis of a similar sample on an older SHRIMP using the SL13 reference zircon gave a similar age-offset (Roberts et al. 1996; Laurie et al. 2016). Wu et al. (2017) find that Kuhfeng Formation (Yangtze Basin, China) samples with 272.95-271.04 Ma CA-TIMS age ranges also come out

1.4% and 1.6% younger when dated by SIMS (a CAMECA 1280, not an ASI SHRIMP). Wu et al. (2017) interpret this as Pb loss which is remediated by chemical abrasion, but which was not avoided by placing ion probe spots on what appear to be undamaged zones of the zircons when viewed in cathodoluminescence or photomicroscopy. The Rowan Formation was deposited between the P2 and P3 Permian glaciations (Metcalfe et al. 2015). In addition to being synchronous, the Rowan Formation and the Kuhfeng Formation are both marine sediments overlain by coal-bearing terrestrial sediments. Perhaps there are diagenetic effects in this sort of environment which would result

in Pb loss from zircon for 3-5 million years after deposition. This would suggest a geological, not analytical problem, which is confounded by comparing chemically abraded ages to SIMS analyses of chemically unabraded zircon.

Sample GA1978296, from the Canning Basin, is 6 Ma, (2.2%) older by SHRIMP than by CA-TIMS. We have no explanation for this, and that result seems to be an inexplicable outlier. Our only suggestion is that relatively few (n=7) zircons were analyzed by SHRIMP, so there may not have been enough measurements to understand the U-Pb systematics of this sample.

In addition to these age outliers, there was a compositional outlier. Sample GA2097849 had much higher U and Th contents than any of the other 35 samples. Its SHRIMP age was more than half a percent younger than the TIMS age, and just outside the uncertainty envelope. This

result is consistent with radiation damage-associated Pb loss which was ameliorated by chemical abrasion, but not avoided by SHRIMP spot placement.

### 4.2.4 Age difference probability density

Figure 5 shows that the distribution of the SHRIMP-TIMS age, expressed in percent, is not Gaussian. Rather, it is bimodal, with approximately two-thirds of the SHRIMP ages up to 0.75% older, and approximately one-third between 0.25 and 1% younger. Including the uncertainties of these differences yields a skewed probability density function. Grouping all 36 samples into a single population requires an excess external (sample to sample) $2\sigma$ error of 0.77%. Added to the lowest SHRIMP uncertainty of 0.47% yields a minimum $2\sigma$ confidence interval of $\pm 1.24\%$. This is within the range of 1-2% uncertainty stated by Schmitt and Vazquez (2017). However, the curve clearly contains

at least two populations (not including the outliers mentioned above), so the mean accuracy is not a particularly meaningful statistic.

Figure 5 shows that the grains where the SHRIMP age is older are predominantly Permian, where the grains with older TIMS ages include most of the Cambrian grains. Comparing age of the grains vs the percent difference (SHRIMP minus TIMS), and shows a reasonable linear regression, with a slope of -2.5% per Ga. Although not perfect, removal of the two >1% outliers described above improves the probability of fit to 0.29. Additionally, the zero intercept is quite close to the age of the Temora 2 reference zircon age. This could be interpreted as

suggesting that there is a time-dependent linear mismatch between SHRIMP and TIMS. However, such an interpretation would be erroneous, for the following reasons.

### 4.2.5 Refutation of linear TIMS-SHRIMP age offset

Most SHRIMP geochronology reported in the scientific literature is used for samples from the Precambrian, not Phanerozoic. Extrapolation of this linear trend into the Precambrian would make the Mesoproterozoic the age mismatch is several percent. This difference is well within

the precision of SHRIMP to detect when comparing $^{207}Pb/^{206}Pb$ vs $^{206}Pb/^{238}U$ ages, but despite thousands of SHRIMP papers being published on this part of the time scale, such a systematic deviation from concordance has not been observed.

Furthermore, there is additional data from most of the sessions used in this study. In addition to the Temora-2 calibration standard and the unknown zircons, 14 of the 16 SHRIMP analytical sessions also contained multiple analyses of the Paleoarchean OG1 zircon (Stern et al. 2009). The slope of -2.4% per Ga predicts that this zircon will have a $^{206}Pb/^{238}U$ age that is 7% younger than the CA-TIMS age. As Figure

2 shows, the offset is more than an order of magnitude smaller. The mean SHRIMP $^{206}Pb/^{238}U$ age from the 16 sessions where OG1 was analysed is lower than the CA-TIMS age, but only by about 0.6%. So no linear age-based calibration offset was present in those analytical sessions.

### 4.2.6 Geological explanation

If chemically abraded material has a fundamentally different U/Pb ratio than chemically unabraded material, then errors in calibration or

systematic biases in the SHRIMP methodology need not be invoked. Rather, it could simply be documenting geologic or crystallographic effects which are remediated by chemical abrasion or avoidable as a result of the excellent single-grain analytical precision of TIMS being able to differentiate zircons from different events only separated in time by a few permil of their age. This explanation is consistent with the

Rowan Formation (GA2031207) results observed above, with the caveat that for some reason the Rowan Formation experienced roughly twice as much Pb loss as the Devonian-Cambrian samples.

For example, zircons from plutonic rocks may have suffered sub-percent Pb loss which chemical abrasion can remedy, but which cannot be avoided by SIMS analytical spot selection. Applying this hypothesis to our data requires us to reclassify it. In Figure 6, we have reclassified the 36 samples of this study, not by age, but by rock type. The "Volcanic" relates to volcanic rocks and ashfall. "Plutonic" is granites, granodiorites, tonalities, porphyries, and other coarse-grained rocks.

All but one of these samples where the SHRIMP age is more than 0.25% older are volcanic, and the TIMS dates may be younger because

the better age-resolution of TIMS on each grain allows sub-million year eruptive edifice-scale antecrysts to be resolved and removed from the best estimate of the eruptive age. This cannot be done for SHRIMP due to the poor counting statistics on individual spots, and the result is ages biased slightly (usually not resolvably) older.

The simplest explanation for the zircons with a SHRIMP age younger than the CA-TIMS age is that the zircons have suffered Pb loss. This population is chiefly made up of plutonic zircons, which are more likely to have accumulated enough radiation damage to have undergone

minor Pb loss.

## 5 Conclusions

The comparison of 36 zircon populations dated by both CA-TIMS and SHRIMP methods shows that the results generally agree to within 1% on most samples. Reporting a SHRIMP age of chemically unabraded zircon with a precision better than ~0.7% increases the chance of that age being different to the CA-ID-TIMS age. The non-Gaussian distribution of the differences makes the relevance of assigning a

Gaussian uncertainty envelope for the accuracy of our SHRIMP results dubious. However, the structure of this mismatch suggests that geologic factors may be part of the disagreement in ages. It is somewhat ironic that what began as a database comparison project has required going back to the actual rocks to find the best explanation.

If confirmed, this geologic explanation has two main implications. Firstly, SIMS geochronology is not the best method in geologic settings where grains may have real differences in crystallization age that are smaller than the precision of a single spot, but larger than the precision

of the final age of the pooled spot values. Permian volcanic rocks which mix eruption-aged zircons grains with zircon crystallized earlier in the history of the igneous complex are one such example. However, the precision of individual spot analyses will depend on the U and Pb content of the zircon, and the size of the analysed volume, with the recognition that as sputter pits deepen to increase volume, effects such as charging of pit walls and ripple formation in the bottom of the crater will eventually disturb the calibration.

Secondly, if much of the analytical error in SHRIMP measurements can be attributed to geologic factors such as Pb loss or antecrysts, then

it is possible that the actual uncertainty in the calibration is less than traditionally stated. The Stern and Amelin (2003) estimate of ~1% was based in part on glass, which should not suffer from these effects. However, that study is almost 20 years old, so refinement of analytical procedures and improvements in SHRIMP manufacturing and installation may have reduced the fundamental uncertainty associated with the calibration equation. Reiners et al. (2017) suggest that excess error should only be added to homogenize dispersed data when a physical explanation cannot be determined. Our hypothesis that geologic factors limit SIMS performance allows the formulation of several testable

hypotheses. Firstly, if Pb loss is a factor, particularly in high U or pre-Permian zircons, further chemical abrasion SIMS experiments can be done to determine the extent to which CA treatment improves or interferes with SIMS U-Th-Pb geochronology. Secondly, methods which

distinguish mixed volcanic zircon, such as $\delta^{18}O$, Hf isotopes, eruption temperature from Ti, trace element composition, and/or $fO_2$ proxies may be applicable to unmixing volcanically mixed zircons prior to U-Pb analysis.

## 6. Author Contributions

CM, SB, and CL produced the new SHRIMP data. J.C. and RF produced the new TIMS data and made tables S2 and S3. SB created the GA geochronology databases used in this study and designed the study with CM. CM recalculated everything as needed, initially interpreted the results and put together most figures and tables. CM, SB, CL, JC and CW wrote the text.

## 7. Acknowledgments

The authors would like to thank the Geoscience Australia mineral separation team for their efforts separating the zircons and preparing the
mounts used in this study, and the Geological Survey of New South Wales for allowing us to publish their TIMS data. We thank the Australian Stratigraphic Units Database (https://asud.ga.gov.au/) for their stratigraphic review tool. The previously published SHRIMP sessions not run by C.L., C.M., or S.B. were run by Emma Chisholm, Andrew Cross, Sharon Jones, Keith Sircombe, and Ian Williams, so we thank them for their efforts. We thank Keith Sircombe for providing figure 1. Further details of the SHRIMP analyses are available via the Geoscience Australia Geochronology Delivery System: http://www.ga.gov.au/geochron-sapub-web/geochronology/shrimp/search.htm .
C.L., C.M., and S.B. publish with the permission of the CEO of Geoscience Australia. All authors thank Yuri Amelin, David Mole, Kathryn Waltenberg, Geoff Fraser, and an anonymous reviewer for reviewing the manuscript. This paper has a Geoscience Australia eCat number of 146201.

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

## 9. Figures

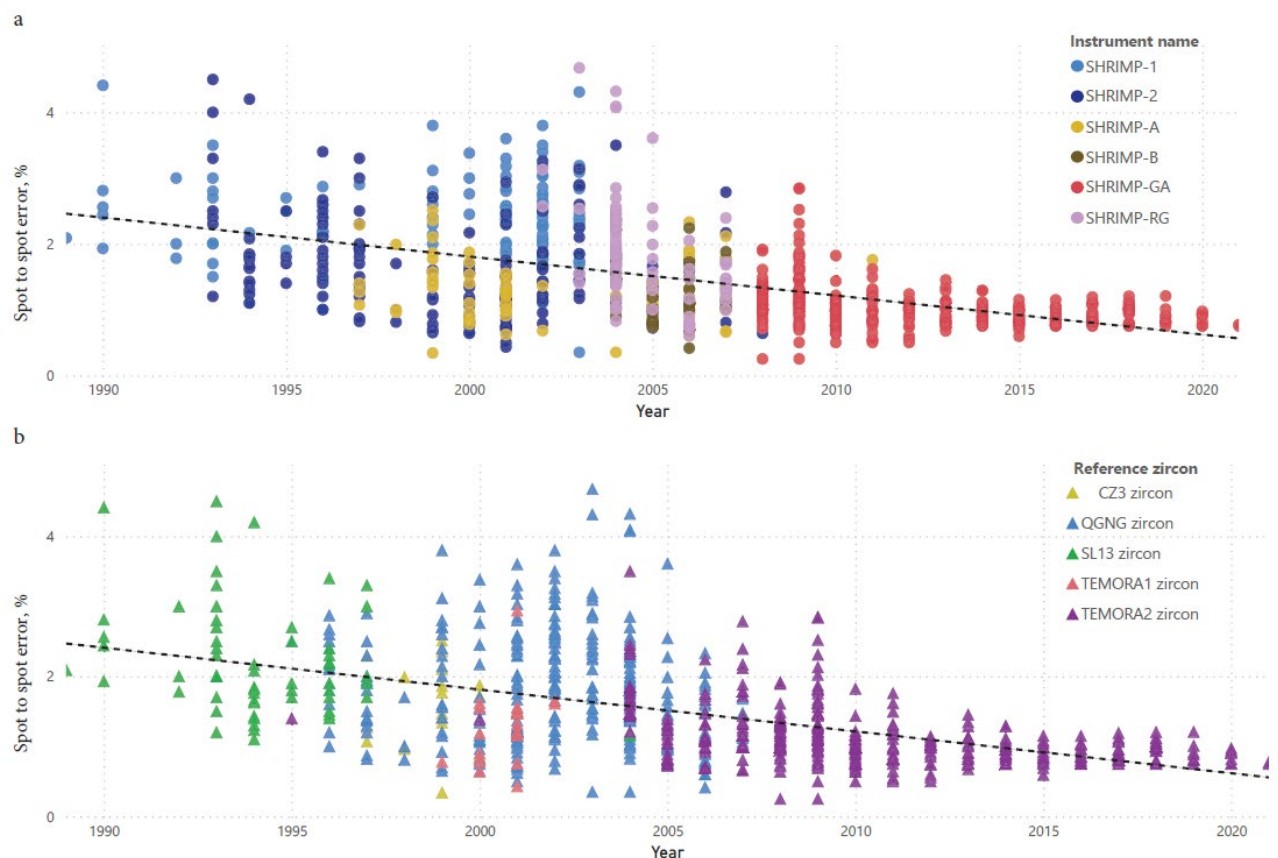

**Figure 1: Graph of session vs spot-to-spot error for all Geoscience Australia SHRIMP sessions producing publishable data through 2021. Figure 1a: data colour-coded by SHRIMP used. SHRIMP 1, 2, and RG are at the Australian National University. SHRIMP A and B are SHRIMP instruments of the SHRIMP 2 design at Curtin University. SHRIMP GA is the SHRIMP 2 instrument at Geoscience Australia. Figure 1b: Data colour-coded by name of primary reference zircon. Results do not include non-GA analyses from university labs. Data in supplementary Table S1.**

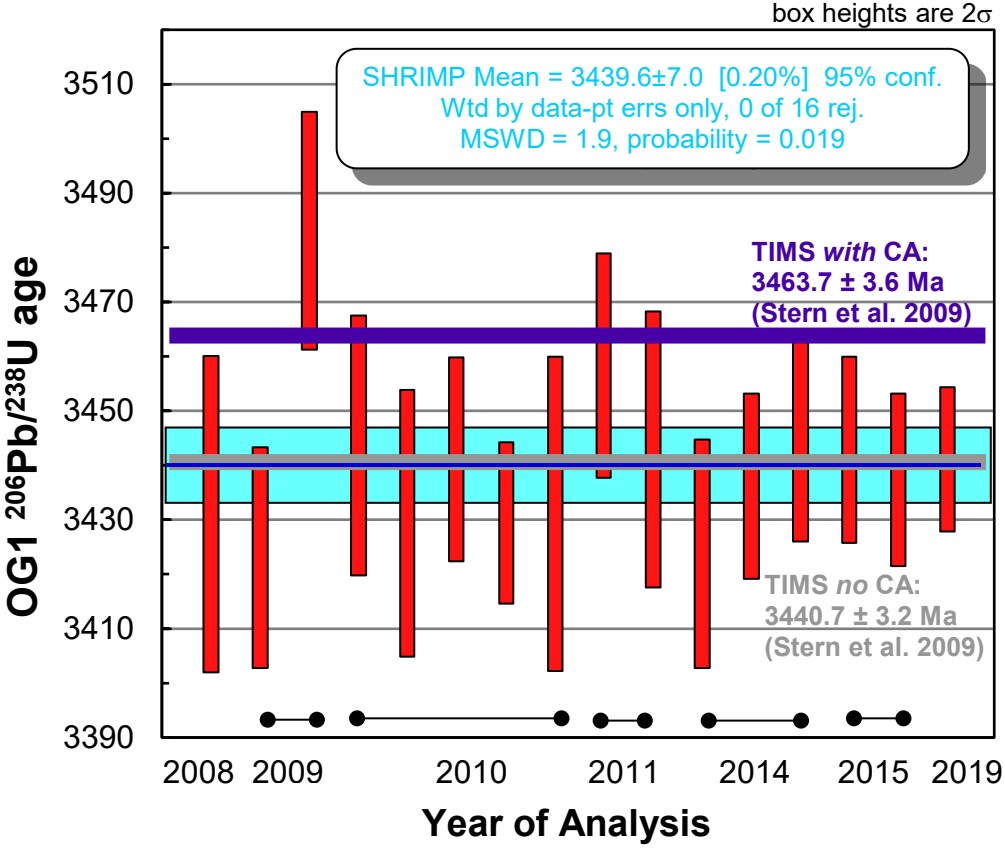


**Figure 2: Weighted mean $^{206}$Pb/$^{238}$U age of OG1 analyses from 14 sessions whose data is included in this paper. The weighted mean is similar to the non-chemical abrasion age, and about 0.5% lower than the chemical abrasion age.**

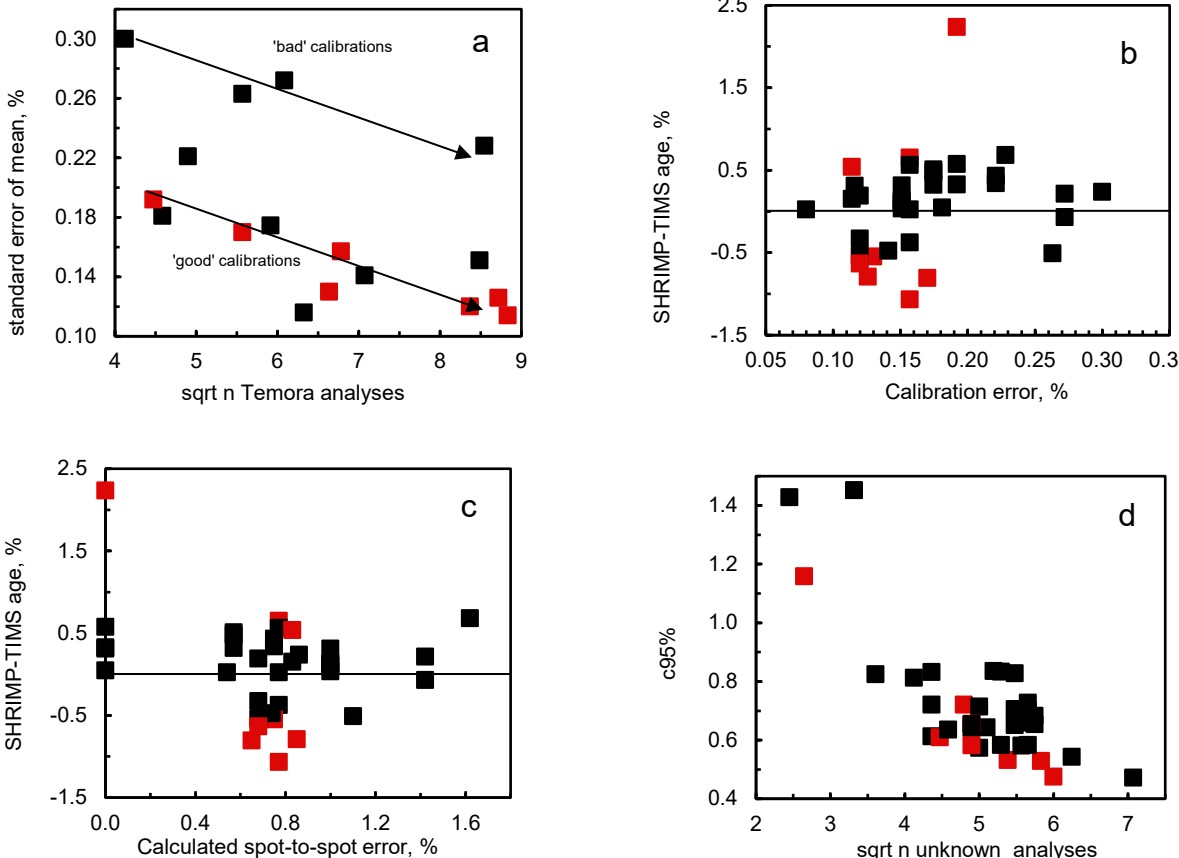


**Figure 3a: Plot of the calibration standard error vs the square root of the number of analyses. Red squares are sessions in which one or more samples had disagreeable TIMS and SHRIMP ages. 3b: Difference in SHRIMP – TIMS age (as a percent of the TIMS age) vs the standard error of the calibration. Red squares are samples where the TIMS and SHRIMP ages disagree. 3c: Difference in**

**SHRIMP – TIMS age (as a percent of the TIMS age) vs the calculated spot-to-spot error of the calibration used for each spot. Red squares are samples where the TIMS and SHRIMP ages disagree. 3d: Total uncertainty for each SHRIMP age vs the square root of the number of unknown analyses. Red squares are samples where the TIMS and SHRIMP ages disagree.**

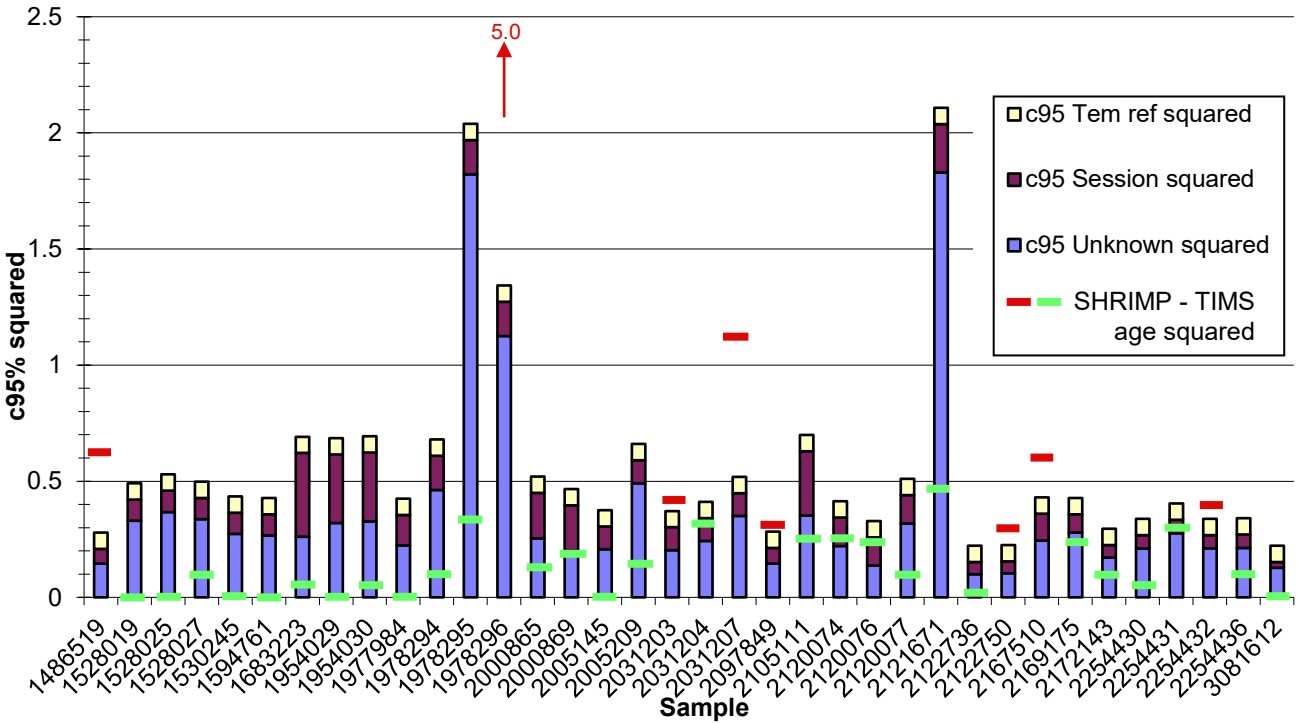

**Figure 4: Error components squared for each unknown sample. The yellow bar is the error component from the reference value of the reference zircon. The maroon bar is the uncertainty derived from the calibration standard error. The blue bar is the remaining error, which is derived from the standard error of the measurements of the unknowns, in addition to a Student's t component. The red bars are the square of the difference between the SHRIMP and TIMS ages for samples where the TIMS and SHRIMP ages disagree. The green bars are for samples where the SHRIMP and TIMS ages are within uncertainty.**

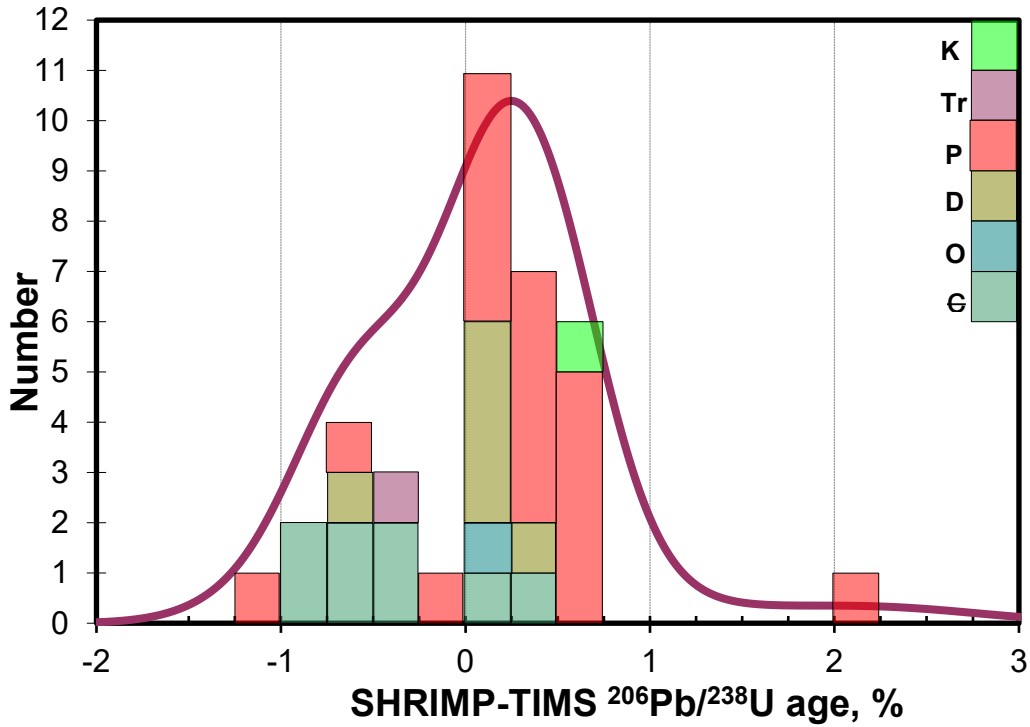

**Figure 5: Histogram and probability density plot for all 36 date comparisons. Histogram is coloured by geologic period.**

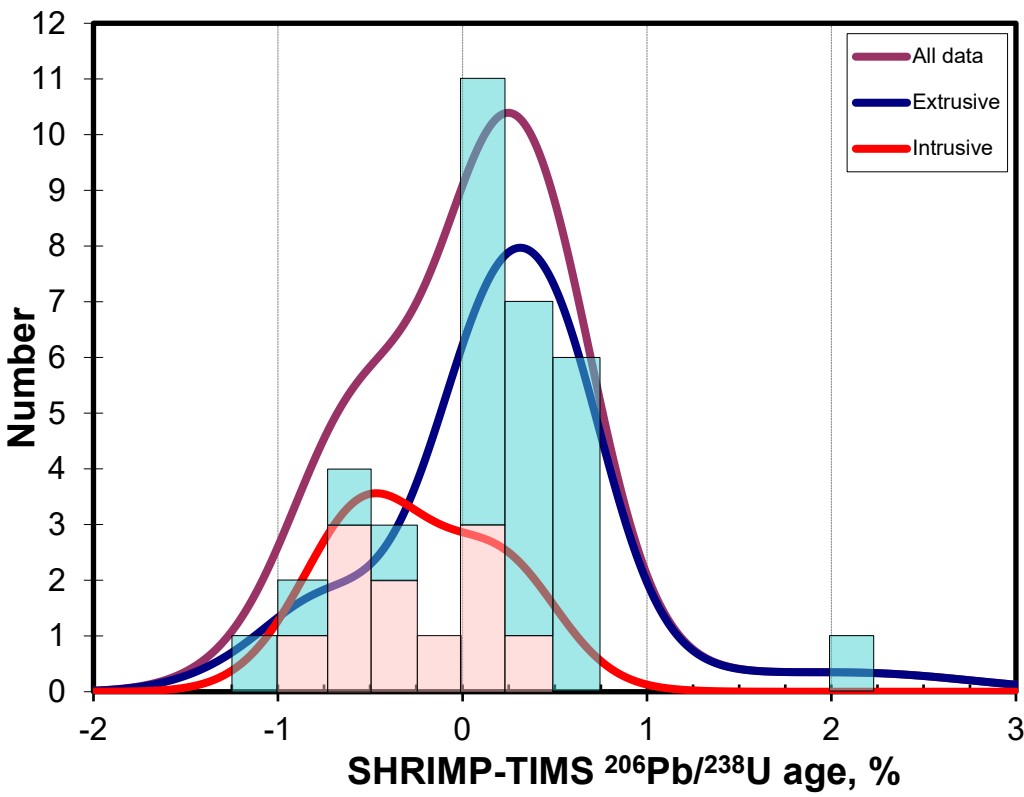

**Figure 6: The same probability curve-histogram combination as Figure 5, but with the histograms recoloured by geologic rock type. Volcanic rocks are blue, plutonic rocks are pink.**


# 10. Tables

**Table 1: Summary information for the 36 samples in the Geoscience Australia database dated using both CA-TIMS and SHRIMP:**

| GA number | Formation | Logged Rock Type | Emplacement Type | SHRIMP reference | TIMS reference | Mount | Session | Median U | Median Th | SHRIMP Year |
|---|---|---|---|---|---|---|---|---|---|---|
| *1486519* | NarrapumelapRoadDaciteMember | Dacite | extrusive | Lewis et al. 2016 | Lewis et al. 2017 † | GA6294 | 150010 | 165 | 178 | 2015 |
| *1528019* | Turondale Fm | Massive volcanic siliciclastic siltstone | extrusive | Bodorkos et al. 2012† | Bodorkos et al. 2012† | GA6075 | 90001 | 190 | 98 | 2009 |
| *1528025* | Upper Merrions Fm | Massive volcanic siliciclastic siltstone | extrusive | Bodorkos et al. 2012† | Bodorkos et al. 2012† | GA6075 | 90001 | 165 | 120 | 2009 |
| *1528027* | Lower Merrions Fm | Massive volcanic siliciclastic siltstone | extrusive | Bodorkos et al. 2012† | Bodorkos et al. 2012† | GA6075 | 90001 | 136 | 91 | 2009 |
| *1530245* | Middle Merions Fm | Porphryritic Volcanic Rock | extrusive | Bodorkos et al. 2012† | Bodorkos et al. 2012† | GA6075 | 90001 | 173 | 108 | 2009 |
| *1594761* | Bull's Camp Volcs | Welded Ignimbrite | extrusive | Bodorkos et al. 2012† | Bodorkos et al. 2012† | GA6075 | 90001 | 336 | 182 | 2009 |
| *1683223* | Dundee Rhyodacite | rhyolite | extrusive | Brownlow&Cross 2010* | Chapman et al. 2022 | GA6058 | 80101 | 375 | 279 | 2008 |
| *1954029* | Parlour Mountain Leucomonzogranite | Granite | intrusive | Cross & Blevin (2010)* | Chapman et al. 2022 | GA6057 | 80104 | 220 | 116 | 2008 |
| *1954030* | Gwydir River adamellite | Granite | intrusive | Cross & Blevin (2010)* | Chapman et al. 2022 | GA6057 | 80104 | 607 | 316 | 2008 |
| *1977984* | Saddington Tonalite | tonalite | intrusive | This Study | This Study | GA6086 | 90038 | 170 | 27 | 2009 |
| *1978294* | Lightjack Fm | Tuff | extrusive | Laurie et al. 2016 | Laurie et al. 2016 | GA6122 | 100044 | 172 | 111 | 2010 |
| *1978295* | Lightjack Fm | Tuff | extrusive | Laurie et al. 2016 | Laurie et al. 2016 | GA6122 | 100044 | 279 | 121 | 2010 |
| *1978296* | Lightjack Fm | Tuff | extrusive | Laurie et al. 2016 | Laurie et al. 2016 | GA6122 | 100044 | 150 | 75 | 2010 |
| *2000865* | Kaloola mbr, Bandana Fm | Tuff | extrusive | Laurie et al. 2016 | Laurie et al. 2016 | GA6112 | 100012 | 339 | 404 | 2010 |
| *2000869* | Kaloola mbr, Bandana Fm | Tuff | extrusive | Laurie et al. 2016 | Laurie et al. 2016 | GA6112 | 100012 | 374 | 242 | 2010 |
| *2005145* | Nalleen Tuff | Tuff | extrusive | Laurie et al. 2016 | Laurie et al. 2016 | GA6113 | 100103 | 305 | 152 | 2010 |

| GA number | Formation | Logged Rock Type | Emplacement Type | SHRIMP reference | TIMS reference | Mount | Session | Median U | Median Th | SHRIMP Year |
|---|---|---|---|---|---|---|---|---|---|---|
| 2005209 | Garie Fm | Felsic Tuff | extrusive | This Study | Metcalfe et al. 2015 | GA6113 | 100103 | 209 | 109 | 2010 |
| 2031203 | Awaba Tuff | Tuff | extrusive | This Study | Metcalfe et al. 2015 | GA6113 | 100103 | 385 | 215 | 2010 |
| 2031204 | Nobby's Tuff | Felsic Tuff | extrusive | This Study | Metcalfe et al. 2015 | GA6113 | 100103 | 304 | 179 | 2010 |
| 2031207 | Rowan fm | Tuff | extrusive | Laurie et al. 2016 | Laurie et al. 2016 | GA6113 | 100103 | 183 | 113 | 2010 |
| 2097849 | Yithan Rhyolite | Porphyritic Rhyolite | intrusive | Bodorkos et al. 2013 | This Study | GA6140 | 100127 | 1179 | 419 | 2010 |
| 2105111 | Emmaville Volcanics | Volcaniclastic rock | extrusive | Cross et al. 2013 | Chapman et al. 2022 | GA6139 | 100128 | 125 | 61 | 2010 |
| 2120074 | Wandsworth Volcanic Group | ignimbrite | extrusive | Chisholm et al. 2014 | Chapman et al. 2022 | GA6152 | 110005 | 272 | 151 | 2011 |
| 2120076 | Wandsworth Volcanic Group | ignimbrite | extrusive | Chisholm et al. 2014 | Chapman et al. 2022 | GA6152 | 110005 | 559 | 328 | 2011 |
| 2120077 | Wandsworth Volcanic Group | crystal tuff | extrusive | Chisholm et al. 2014 | Chapman et al. 2022 | GA6152 | 110005 | 340 | 170 | 2011 |
| 2121671 | unknown | calcilutite | extrusive | This Study | This Study | GA6155 | 110011 | 89 | 63 | 2011 |
| 2122736 | Tinowon fm | Tuff | extrusive | Laurie et al. 2016 | Laurie et al. 2016 | GA6169 | 110088 | 395 | 221 | 2011 |
| 2122750 | Kaloola mbr, Bandana Fm | Tuff | extrusive | Laurie et al. 2016 | Laurie et al. 2016 | GA6169 | 110088 | 292 | 269 | 2011 |
| 2167510 | 'Victorporphyry' | Porphyry | intrusive | Lewis et al. 2015 | Lewis et al. 2017 † | GA6268 | 140028 | 143 | 126 | 2014 |
| 2169175 | 'Victorporphyry' | Porphyry | intrusive | Lewis et al. 2015 | Lewis et al. 2017 † | GA6272 | 140035 | 147 | 159 | 2014 |
| 2172143 | Unnamedporphyryhostingmolybdenite | Porphyry | intrusive | Lewis et al. 2016 | Lewis et al. 2017 † | GA6282 | 140082 | 97 | 61 | 2014 |
| 2254430 | BushyCreekGranodiorite | Granodiorite | intrusive | Lewis et al. 2016 | Lewis et al. 2017 † | GA6302 | 150060 | 112 | 71 | 2015 |
| 2254431 | BuckeranDiorite | Diorite | intrusive | Lewis et al. 2016 | Lewis et al. 2017 † | GA6302 | 150060 | 257 | 365 | 2015 |
| 2254432 | BuckeranDiorite | Diorite | intrusive | Lewis et al. 2016 | Lewis et al. 2017 † | GA6302 | 150060 | 158 | 238 | 2015 |

| GA number | Formation | Logged Rock Type | Emplace ment Type | SHRIMP reference | TIMS reference | Mount | Session | Median U | Median Th | SHRIMP Year |
|---|---|---|---|---|---|---|---|---|---|---|
| *2254436* | BushyCreekGrano diorite | Granodiorite | intrusive | Lewis et al. 2016 | Lewis et al. 2017 † | GA6302 | 150060 | 91 | 52 | 2015 |
| *3081612* | Emmaville Volcanics | Volcaniclastic rock | extrusive | Jones et al. in prep | Chapman et al. 2022 | GA6430 | 190057 | 445 | 252 | 2019 |

*\* Reprocessed using SQUID 2.5*

*† Full data in this paper*


**Table 2: Summary information for the 16 SHRIMP analytical sessions in which these data were collected. N Tem= number of Temora-2 analyses as the primary reference material; n Tem Excl- number of excluded standard spots; 1σ er f mean- uncertainty of calibration; spot to spot er- additional error for each spot to give MSWD=1; Temora common Pb isotope- common Pb isotope used for primary reference material; Unknown common Pb isotope- common isotope of Pb used for unknowns in session; n OG1 number of OG1 analyses used to determine $^{207}Pb/^{206}Pb$ fractionation; OG1 $^{206}Pb/^{238}U$ age- the U-Pb age (not the $^{207}Pb/^{206}Pb$ age) of the $^{207}Pb/^{206}Pb$ standard material; OG1 c95% abs WITH ref er of tem mean- uncertainty on OG1 $^{206}Pb/^{238}U$ age:**


| session | n Tem | n Tem excluded | 1σ er of mean, % | spot –to spot er % | $^{204}Pb$ over-counts/s from $^{207}Pb$ | over-count 95% conf | Tem common Pb isotope | Unknown common Pb isotope | n OG1 | n OG1 excl | OG1 $^{206}Pb/^{238}U$ age | OG1 c95% abs WITH ref er of tem mean |
|---|---|---|---|---|---|---|---|---|---|---|---|---|
| 80101 | 17 | 0 | 0.30 | 0.86 | -0.01 | 0.03 | 207 | 207 | 0 | | | |
| 80104 | 37 | 0 | 0.27 | 1.42 | -0.01 | 0.02 | 207 | 207 | 16 | 0 | 3430.9 | 29.1 |
| 90001 | 72 | 0 | 0.15 | 1.00 | -0.01 | 0.02 | 204 | 204 | 18 | 0 | 3422.9 | 20.3 |
| 90038 | 21 | 0 | 0.18 | 0.75* (0) | 0.01 | 0.02 | 204 | 204 | 10 | 0 | 3483.0 | 21.8 |
| 100012 | 24 | 0 | 0.22 | 0.75 | 0.00 | 0.02 | 207 | 207 | 11 | 0 | 3443.4 | 23.8 |
| 100044 | 20 | 0 | 0.19 | 0.75* (0) | 0.01 | 0.03 | 207 | 207 | 8 | 0 | 3429.2 | 24.6 |
| 100103 | 46 | 0 | 0.16 | 0.77 | 0.01 | 0.01 | 207 | 207 | 20 | 0 | 3441.0 | 18.7 |
| 100127 | 44 | 0 | 0.13 | 0.75 | -0.01 | 0.02 | 204 | 204 | 21 | 0 | 3429.2 | 14.9 |
| 100128 | 31 | 0 | 0.26 | 1.10 | 0.01 | 0.02 | 204 | 204 | 14 | 1 | 3430.9 | 28.9 |
| 110005 | 36 | 1 | 0.17 | 1.00* (0.57) | 0.02 | 0.02 | 204 | 204 | 26 | 0 | 3458.2 | 20.6 |
| 110011 | 78 | 5 | 0.23 | 1.62 | 0.04 | 0.02 | 204 | 207 | 31 | 0 | 3442.7 | 25.4 |
| 110088 | 79 | 1 | 0.11 | 0.83 | 0.02 | 0.02 | 207 | 207 | 0 | | | |
| 140028 | 31 | 0 | 0.17 | 0.75* (0.65) | 0.02 | 0.02 | 204 | 204 | 14 | 0 | 3423.6 | 21.0 |
| 140035 | 51 | 1 | 0.14 | 0.75* (0.74) | 0.01 | 0.01 | 204 | 204 | 11 | 0 | 3436.1 | 17.1 |
| 140082 | 40 | 0 | 0.12 | 0.75* (0) | 0.02 | 0.01 | 204 | 204 | 16 | 0 | 3444.3 | 18.4 |
| 150010 | 76 | 0 | 0.13 | 0.85 | 0.00 | 0.01 | 204 | 204 | 27 | 1 | 3442.8 | 17.1 |
| 150060 | 70 | 0 | 0.12 | 0.75* (0.68) | 0.02 | 0.01 | 204 | 204 | 28 | 0 | 3437.2 | 15.9 |
| 190057 | 70 | 0 | 0.08 | 0.75* (0.56) | -0.01 | 0.01 | 207 | 207 | 20 | 0 | 3440.9 | 13.3 |

**Table 3: Summary of SHRIMP – CA-TIMS comparison. TIMS older- antecryst zircons analysed; TIMS grouped for age- n of zircons used in age calculation; TIMS younger- younger ungrouped zircons: SHRIMP older- antecryst zircons analysed; SHRIMP grouped for age- n of zircons used in age calculation; SHRIMP younger- younger ungrouped zircons; Diff (Ma) difference in age in Ma; Diff %- difference in age in % of the TIMS age; Diffc95% difference in age in numbers of SHRIMP 95% confidence intervals.**


| GA Sample No | TIMS older | TIMS group-ed for age | TIMS young-er | SHRIMP older | SHRIMP grouped for age | SHRIMP younger | CA-TIMS $^{206}Pb/^{238}U$ age | ± c95% (random + tracer) | SHRIMP $^{206}Pb/^{238}U$ age | ± c95% (random + session + reference) | Diff (Ma) | Diff % | Diff c95% |
|---|---|---|---|---|---|---|---|---|---|---|---|---|---|
| 1486519 | 1 | 7 | 0 | 2 | 34 | 0 | 507.21 | 0.39 | 503.2 | 2.7 | -4.01 | -0.79 | -1.51 |
| 1528019 | 0 | 6 | 0 | 0 | 32 | 0 | 415.56 | 0.51 | 416 | 2.9 | 0.44 | 0.11 | 0.15 |
| 1528025 | 2 | 3 | 0 | 0 | 32 | 0 | 412.73 | 0.96 | 413.3 | 3.0 | 0.57 | 0.14 | 0.19 |
| 1528027 | 2 | 4 | 0 | 1 | 30 | 1 | 411.71 | 0.89 | 413 | 2.9 | 1.29 | 0.31 | 0.44 |
| 1530245 | 0 | 4 | 1 | 1 | 31 | 0 | 413.76 | 0.76 | 414.2 | 2.7 | 0.44 | 0.11 | 0.16 |
| 1594761 | 0 | 3 | 2 | 0 | 33 | 0 | 417.75 | 0.88 | 417.9 | 2.7 | 0.15 | 0.04 | 0.05 |
| 1683223 | 0 | 6 | 0 | 0 | 19 | 0 | 253.10 | 0.15 | 253.7 | 2.1 | 0.6 | 0.24 | 0.28 |
| 1954029 | 0 | 6 | 0 | 0 | 30 | 0 | 255.08 | 0.16 | 254.9 | 2.1 | -0.18 | -0.07 | -0.09 |
| 1954030 | 3 | 3 | 0 | 2 | 28 | 0 | 252.76 | 0.26 | 253.3 | 2.1 | 0.54 | 0.21 | 0.26 |
| 1977984 | 0 | 7 | 0 | 2 | 30 | 0 | 487.07 | 0.70 | 487.3 | 3.2 | 0.23 | 0.05 | 0.07 |
| 1978294 | 0 | 7 | 0 | 10 | 13 | 3 | 268.02 | 0.16 | 268.9 | 2.2 | 0.88 | 0.33 | 0.40 |
| 1978295 | 0 | 8 | 0 | 0 | 6 | 4 | 269.25 | 0.10 | 270.8 | 3.9 | 1.55 | 0.58 | 0.40 |
| 1978296 | 1 | 8 | 0 | 3 | 7 | 1 | 268.79 | 0.14 | 274.8 | 3.2 | 6.01 | 2.24 | 1.89 |
| 2000865 | 0 | 8 | 0 | 0 | 19 | 1 | 252.64 | 0.10 | 253.5 | 1.8 | 0.86 | 0.34 | 0.47 |
| 2000869 | 1 | 12 | 0 | 1 | 33 | 2 | 253.11 | 0.08 | 254.2 | 1.7 | 1.09 | 0.43 | 0.63 |
| 2005145 | 1 | 7 | 2 | 1 | 19 | 1 | 253.14 | 0.09 | 253.2 | 1.6 | 0.06 | 0.02 | 0.04 |
| 2005209 | 0 | 8 | 0 | 4 | 17 | 4 | 248.23 | 0.19 | 247.3 | 2.0 | -0.93 | -0.37 | -0.46 |
| 2031203 | 1 | 7 | 0 | 2 | 20 | 1 | 253.25 | 0.13 | 254.9 | 1.6 | 1.65 | 0.65 | 1.06 |
| 2031204 | 1 | 7 | 0 | 1 | 24 | 0 | 255.26 | 0.14 | 256.7 | 1.6 | 1.44 | 0.56 | 0.87 |
| 2031207 | 2 | 6 | 0 | 1 | 23 | 4 | 271.60 | 0.13 | 268.7 | 1.9 | -2.9 | -1.07 | -1.50 |
| 2097849 | 1 | 7 | 0 | 6 | 29 | 6 | 414.27 | 0.26 | 412.0 | 2.2 | -2.27 | -0.55 | -1.04 |
| 2105111 | 1 | 5 | 0 | 0 | 27 | 0 | 253.59 | 0.17 | 252.3 | 2.1 | -1.29 | -0.51 | -0.61 |
| 2120074 | 2 | 4 | 0 | 0 | 26 | 0 | 254.70 | 0.19 | 256.0 | 1.6 | 1.3 | 0.51 | 0.79 |
| 2120076 | 0 | 6 | 0 | 0 | 25 | 0 | 254.58 | 0.15 | 255.8 | 1.5 | 1.22 | 0.48 | 0.83 |

| GA Sample No | TIMS older | TIMS grouped for age | TIMS younger | SHRIMP older | SHRIMP grouped for age | SHRIMP younger | CA-TIMS $^{206}Pb/^{238}U$ age | ± c95% (random + tracer) | SHRIMP $^{206}Pb/^{238}U$ age | ± c95% (random + session + reference) | Diff (Ma) | Diff % | Diff c95% |
|---|---|---|---|---|---|---|---|---|---|---|---|---|---|
| 2120077 | 1 | 5 | 0 | 0 | 25 | 0 | 255.48 | 0.17 | 256.3 | 1.8 | 0.82 | 0.32 | 0.45 |
| 2121671 | 2 | 7 | 0 | 4 | 11 | 1 | 139.15 | 0.09 | 140.1 | 2.0 | 0.95 | 0.68 | 0.47 |
| 2122736 | 6 | 4 | 0 | 0 | 50 | 1 | 256.01 | 0.11 | 256.4 | 1.2 | 0.39 | 0.15 | 0.32 |
| 2122750 | 1 | 8 | 0 | 0 | 36 | 0 | 252.54 | 0.09 | 253.9 | 1.2 | 1.36 | 0.54 | 1.13 |
| 2167510 | 1 | 5 | 0 | 2 | 24 | 1 | 504.17 | 0.31 | 500.1 | 3.3 | -4.07 | -0.81 | -1.24 |
| 2169175 | 0 | 6 | 0 | 2 | 24 | 1 | 503.80 | 0.29 | 501.4 | 3.3 | -2.4 | -0.48 | -0.73 |
| 2172143 | 0 | 8 | 0 | 0 | 39 | 1 | 504.53 | 0.31 | 506.1 | 2.7 | 1.57 | 0.31 | 0.57 |
| 2254430 | 1 | 6 | 0 | 3 | 31 | 2 | 501.55 | 0.31 | 502.5 | 2.9 | 0.95 | 0.19 | 0.33 |
| 2254431 | 0 | 8 | 0 | 2 | 21 | 2 | 504.83 | 0.30 | 502.1 | 3.2 | -2.73 | -0.54 | -0.86 |
| 2254432 | 0 | 6 | 1 | 0 | 24 | 1 | 505.00 | 0.36 | 501.8 | 2.9 | -3.2 | -0.63 | -1.10 |
| 2254436 | 0 | 6 | 0 | 4 | 28 | 2 | 501.65 | 0.29 | 500.0 | 2.9 | -1.65 | -0.33 | -0.57 |
| 3081612 | 2 | 5 | 0 | 0 | 32 | 0 | 256.04 | 0.14 | 256.1 | 1.2 | 0.06 | 0.02 | 0.05 |

| | | | | | | | | | | Median Absolute Deviation | 0.8 | 0.3 | 0.3 |
| | | | | | | | | | | Median | 0.49 | 0.15 | 0.22 |
| | | | | | | | | | | Simple Mean | 0.022 | 0.095 | 0.073 |

**Table S-1: Calibration and session information for all Geoscience Australia SHRIMP zircon sessions (1989-2021) (Electronic).**

**Table S-2: Grain-by-Grain CA-TIMS data from Boise State University (Electronic).**

**Table S-3: Grain-by-Grain CA-TIMS data from University of British Columbia (Electronic).**

**Table S-4: Spot by spot data for new SHRIMP results from Geoscience Australia SHRIMP (Electronic).**