# Peer review of "Examination of the accuracy of SHRIMP U-Pb geochronology based on samples dated by both SHRIMP and CA-TIMS"

_Geochronology, 2022_

## Author Response (AR1)

Reply to referee reports for Magee et al. gchron-2022-20:

We thank both reviewers for their insightful reviews. In response, we would like to take this opportunity to improve the structure of the paper. The source of dispersion (excess to counting stats) in U-Pb SIMS data has been debated for decades; this debate can most succinctly be summarized by the arguments of Black and Jagondinski (2003) on one hand, who say that an excess error term needs to be applied if the source cannot be identified, and Compston (2000), who argues that all the scatter can be traced to bad zircons. We didn't include either paper in our introduction as the actual data they were discussing (multi-grain pre-CA TIMS aliquots vs SHRIMP 1 data) are obsolete, but the ideas are still current. Indeed, and Black and Jagodzinski (2003) say in their subsection "The Way Ahead":

> " There are two alternative courses of action to adopt once it is accepted that uncertainties exceeding those predicted from counting statistics can be generated as part of the SHRIMP analytical process. The first is to empirically quantify the magnitude of variation by means of replicate analyses, and then to use SHRIMP only for those projects where such variation (e.g. 1–2%) is acceptable. The other approach is to delve as deeply and objectively as possible into the various sources of uncertainty in SHRIMP dating, so that they can be identified, understood and ultimately either minimised or removed altogether."

Moving our treatment of uncertainty from the intro to the methods (as reviewer 2 suggests) will allow us to streamline the introduction to highlight how we can now examine both calibration issues and inhomogenous natural sample issues using our data of doubly dated zircons.

As both reviewers recommend removing figure 3 and minimizing the associated analysis, we can replace this with discussion about the calibration. In the current manuscript we skip that analysis and go straight to looking at natural mineral-based explanations, which is jarring when you consider how much of the introduction is spent explaining the calibration. We have re-examined the calibration-related data enough to know that there is information there to discuss, and this would make the paper more interesting and help tease out the competing factors of calibration quality vs geological accuracy.

This leads us to the relevance and usefulness of considering sub-percent SIMS data. It should not be surprising that most of our data are reported with sub-percent precision, as most current SIMS U-Pb data falls into this category. In both rounds of the recent G-Chron U-Pb proficiency testing (Webb et al., submitted), the reported median 2sig uncertainty for the $^{206}Pb/^{238}U$ SIMS age was 0.5% in round 1 and 0.6% in round 2. As our dataset contains many samples in this range, it is worth considering how SIMS data in that range of precision compares to CA-ID-TIMS.

This leads us to the questions about the source and application of uncertainty in central to the comments from reviewer 1 about the statistical validity of our data analysis.

Using the data as reported, even with the exclusion of the outliers discussed in section 4.2.2, the weighted mean of the reported data has an MSWD of 1.9 and a probability of fit of 0.002, indicating that it is not a homogenous population. Obviously the probability of fit can be increased by arbitrarily increasing the uncertainties of each age- increasing them to 2% give an MSWD of 0.35 and a 100% probability of fit, for example. But as Reiners et al. (2017) suggest in chapter 4, hiding dispersion by the use of excess error should only be done if a physical explanation cannot be found; this is why it is important to consider explanations related to the physical samples. As far as our hypothesis that the data contains two populations is concerned, we would like to point out that the mixture-modelling approach of Sambridge & Compston (1994) shows that the statistically most likely population split is similar to that derived from dividing the samples into intrusive and extrusive rocks.

Obviously we cannot accommodate Reviewer 1's request to both cull data and have a larger dataset. One result of compiling this data, which was first done in the late 2010's, is that it tells us which samples are not likely to yield geologically useful ages from SHRIMP analyses, and should be sent straight to a CA-ID-TIMS lab. As a result, the incidence of double dating has been reduced, and there is only one additional doubly dated sample which has appeared since 2016. As it wasn't available until the manuscript was in internal review, and it is yet another outcrop of the Emmaville Volcanics, which are already over-represented relative to the rest of the Australian continent in this study, we didn't add it in between reviewers. However, we have included it in the final paper. As for culling data, we feel that a discussion of the calibration behaviour and quality (see above) will help put various outliers in context.

We appreciate reviewer 1's suggestion of Burgess et al. (2019), and agree that it is a great paper on Pleistocene tuff dating. However, the papers on Permian tuffs cited by us which describe the zircons discussed in this paper are more representative of the problems we have in these rocks. For example, figure 12 in Metcalfe et al. (2015), shows that most of the samples in this drill core which have been analysed by CA-ID-TIMS contain zircons crystals which predate the eruption age. A particularly striking example is the second lowest tuff, GA2122738, with an eruption age of 254.34 +/- 0.08 Ma. The next three tuffs above this in the drill core contain inherited (or contaminant) zircons whose age is within uncertainty of the GA2122738 eruption age. These grains are excluded from the weighted mean eruption age in each case because the TIMS is precise enough to identify them.

The uppermost tuff in this sequence (GA2122750) is one of the SHRIMP analyses in which the SHRIMP age is older than, and not within uncertainty of, the CA-ID-TIMS age. However, the old outlier grain in the CA-ID-TIMS analysis of GA2122750 is within error of the SHRIMP age, as are the next six tuffs lower down in the drill hole. The 2sig uncertainty on each individual SHRIMP spot for this sample is on the order of 5 Ma, so distinguishing spots on antecrysts instead of eruption-age zircons by U-Pb date alone is impossible, allowing accidental antecryst analyses to bias the SHRIMP ages older. .

Point by point response to reviewer one (original comment *in italics*):

*Line 23: This apparent bimodality needs to be statistically verified.*

Line 23. The issue of statistical probability for bimodality is addressed above.

*Line 25: "better single-grain age-resolution of TIMS" = this is a bit awkward to read, as the integration of multiple age domains is the main drawback of TIMS. I also doubt if CA-TIMS can resolve genuine pre-eruptive zircon crystallization in the same magma system (= antecrysts) in the age range presented. Even if it did, what would this mean for dating a geological event such as deposition of a tephra (see Keller et al., 2018)?*

Line 25: Addressed above with regards to zircon recycling shown in Metcalfe et al. (2015)

*Line 298: is not included*

298: Changed.

*Line 310: This section is repetitive and tedious to read; this can be condensed to summarizing the main points in a table. In fact, I think section 4.1 which is presently in the discussion, should be presented as the main result.*

310: While this is repetitive to read, it makes these results much more searchable and easy to find. Several of these TIMS results are close enough to various stage or period boundaries to be interesting

to non-SIMS people, and putting the names and ages in the text together makes them easier for both humans and machines to find.

*Line 402: Pb-loss after 3-5 million years seems highly speculative, and not supported by any experimental data on zircon interaction with fluids. As U abundances are not discussed, there is no way to gauge timescales for metamictization, but this is something that the authors should look into and add to the presentation.*

402: We agree that this seems unusual, and yet this phenomenon first observed by Wu et al. (2017) is also present here, so we feel it should be reported. Repeatable unexplicable results are important for the advancement of science.

*Line 405: Yes, I totally agree that this is not statistically robust.*

405: agreed.

*Line 408: I disagree that the shape of the distribution has been assessed in a statistically robust way.*

408: Addressed in the text above

*Line 416. Not sure if "p-hacking" is an adequate term; in any case, it has a negative connotation.*

416: Changed.

*Line 456: Why would this be more likely? There seem to be some underlying assumptions here that should be explicitly stated.*

456: Rectification of minor Pb loss is the whole point of chemical abrasion, so it is the most sensible hypothesis for natural materials being apparently younger, based on the last 15 years of TIMS work.

*Lien 457: This sentence is awkward: What are natural ages? What are chemically abraded ages?*

457: Changed.

*Line 465: I am not convinced that this is a valid interpretation.*

465: The reviewer is either being stubborn or obtuse. SIMS ages with reported precision of less than 0.7% are more than twice as likely to disagree with the CA-ID-TIMS ages than those which have larger uncertainties. This point directly contradicts his main text assertion that reported uncertainties for SIMS analyses should be larger.

*Line 469: "SIMS geochronology is not the best method in geologic settings where grains may have real differences in crystallization age that are smaller than the precision of a single spot, but larger than the precision of the final age of the pooled spot values." I don't agree with this statement, as a bulk method will create artificially small uncertainties for an age that may not have any geological significance (see discussion in Keller et al., 2018, and elsewhere).*

469: This comment is either illogical or irrelevant. The reviewer states "a bulk method will create artificially small uncertainties for an age that may not have any geological significance (see discussion in Keller et al., 2018, and elsewhere)" We agree (see arguments in Ickert et. al 2015, of which CM was an author). But that has no bearing on our statement, that "SIMS geochronology is not the best method in geologic settings where grains may have real differences in crystallization age." This is particularly the case in the ashfall deposits of the Eastern Australian coalfields, where sub-million year ashfall deposits often contain (and sometimes only contain) zircons whose age agrees with the

eruption ages of lower ashfall units in the same drillcore. This is a specific geological issue relating to Australian Permian volcaniclastics, which are half of our total samples. We invoke geological explanations for a specific reason- that the previously identified geologic issues which have been confirmed in the studies which provide most of our TIMS data (Chapman et al. (2022), Laurie et al. (2016), and Metcalf et al. 2015)) are relevant to both TIMS and SIMS, as they are the same physical zircons.

*Line 478: "improvements in SHRIMP manufacturing and installation may have reduced the fundamental uncertainty associated with the calibration equation" "May have" reads awkward; the data and interpretation in this paper at least do not support this.*

478: Addressed on our revised introduction

*Fig. 2: The PDF is based on assigned uncertainties that may or may not be adequate (see comment to Fig. 4)*

Figure 2: They are based on the uncertainties which are calculated. As mentioned in our text, assigning artificially large uncertainties could homogenize the data, but that would make trying to understand, analyse, and discuss the source of the overdispersion impossible, which is the whole point of the paper;.

*Fig. 3: I would omit this plot; the fit has a probability of only 0.003, the slope generated is probably an artifact of the data selection, and the results for OG1 show that this relation is invalid (including younger reference zircon would probably also confirm this). "Cherry picking" and "p-hacking": why even go there?*

Figure 3. Agreed.

*Fig. 4: MSWD and probability of fit suggest that there is overdispersion/underestimation of uncertainties for the SHRIMP results.*

Figure 4. This figure and its interpretation have been reorganized according to the suggestions of Reviewer 2, and the point raised here is discussed.

*Fig. 5: Statistical testing of the difference/equivalence of both distributions would be required to demonstrate that this distinction is significant (e.g., using a Kolmogorov—Smirnov comparison).*

Figure 5. See above for discussion of statistics.

*Lines 45-46. To what extent the study of Jeon and Whitehouse (2014) is relevant to SHRIMP usage? Is the difference between the Cameca and SHRIMP design sufficiently big to make their results inapplicable to SHRIMPs?*

45-46 Fair point, but we wanted to emphasize this was the only study done on \*any\* SIMS platform since the early 00s.

*Lines 78-80. The qiestion here is when the uncertainty of calibration is applied: before or after averaging the sample spot analyses. I think the latter is correct approach, as it prevents artificial uncertainty reduction due to repetition. The same dilemma exists, and is widely acknowledged, in Ar-Ar geochronology.*

78-80 This is a fair question, but for most SIMS data reduction it has been done before averaging (Ludwig 2009). What the consequences of this are, and whether the spot to spot is an appropriate precision floor is one of the things we are investigating.

*Lines 85-91. Make it clear that you talk about age (or 238U/206Pb) standards here. Temora-2 is indeed superior to SL13 as an age standard. However, as a concentration standard SL13 is significantly better than any Temora zircon. This is why modern SHRIMP studies use both standards together, each to its strength.*

85-91. This is a fair point, but this paper deals exclusively with the 206Pb/238U system. Cancentration reference material sare only used for the indicative U and Th median concentrations shown in table 1.

*Line 102. Another good paper on the basics of chemical abrasion is Mattinson (2011) Extending the Krogh legacy: development of the CA–TIMS method for zircon U–Pb geochronology, Canadian Journal of Earth Sciences v.48, pp.95-105 (the special volume dedicated to memory of Tom Krogh).*

102. Agreed.

*Lines 131-134. There are pros and contras in using samples collected for geological problem solving vs. dedicated natural standards. The downside of the "in the wild" approach here is that the "geological" uncertainty is typically greater compared to using natural reference materials (i.e., the best preserved and most homogeneous minerals). It would be interesting to discuss this topic in more detail.*

131-134 Our rewrite expands on "in the wild" geologic factors.

*Lines 138-139. Consider recalculating SHRIMP ages using the age of Temora zircon reported by Schaltegger et al. (2021) JAAS DOI: 10.1039/d1ja00116g. The difference is likely to be small, but this would still make the data a bit more accurate.*

138-139. We are using the Temora reference age from non-chemically abraded Temora, partially because we don't yet know how comparable CA ages are for untreated material (that's one of the aims of this study), but mostly because we are using archived data that predates Schaltegger et al. 2021. In none of these samples were the Temora-2 zircons on the mount chemically abraded, so we use a non-chemically abraded reference value for that reference material. Changing the reference value would have the effect of moving the entire data set up or down the time scale, so it would reduce the outlier on one side at the expense of increasing those on the other.

*Line 147. Metcalfe (a typo). The same in line 302.*

147 Fixed

*Lines 233-234. Sounds like zircon solutions were put into anion exchange separation in 6M HCl. This does not make sense, and is not consistent with the Krogh (1973) chemistry or its later adaptations. Pb does not stick to the resin in this medium.*

*Line 235. Eluted together in what medium?*

233-235 Methods corrected.

*Lines 373-374. Strictly speaking, you should use a quadratic sum of both confidence intervals. The difference from using SHRIMP confidence interval would be small, however.*

373-374. We checked, and in no case does including the TIMS confidence interval bridge the gap in samples which disagree. So for simplicity the SHRIMP interval (which is on average about 8 times larger, meaning the square is 64 times larger) is used. In our new figure 3, the TIMS uncertainty, although variable from sample to sample, is generally around the width of the red or green indicator line.

*Section 4.1. I think it should be part of "Results" rather than "Discussion".*

Section 4.1 Rearranged in the rewrite

*Lines 430-434. Extrapolation of the trend defined from a narrow spread of values (in this case, the age) to a much wider range, such as shown in Fig. 3, is usually unreasonable. At the very least, show the uncertainty envelope for the entire range from 0 to 3500 Ma, not just ~100-500 Ma as it is done now. This will immediately and clearly show how much significance does this slope really have.*

430. Figure 3 removed

*Lines 424-438. About OG1. All data that are discussed in a paper must be introduced in "Results". It is not permissible to introduce any new data in the "Discussion" section. Hence add a brief section about OG1 to the "Results".*

424: Rearranged in the rewrite

*Lines 434-435. Add references to support this statement.*

434 Done.

*Lines 436-438. Consider the distribution of radiation damage in understanding how chemical abrasion works. The damage can vary from individual recoil tracks (of ca. 100-150 nm) to U-rich bands in oscillatory zoning (microns or wider). Dissolution of individual recoil tracks would not make any visible changes in the zircon, but would impact the U-Pb system.*

436 Noted.

*Lines 455-456. The extent of radiation damage can be estimated with the data available in this study, without any additional measurements (especially if we consider U and Th but ignore Sm, which may be a sensible approach). It would be good to seach for any possible correlations between radiation damage and U-Pb systematics.*

455 Incorporated in rewrite

                                                                                           **Insert document title here**

References cited:

Black, L. P., Jagodzinski, E. A. Importance of establishing sources of uncertainty for the derivation of reliable SHRIMP ages, Australian Journal of Earth Sciences, 50:4, 503-512, 2003. DOI: 10.1046/j.1440-0952.2003.01007.x

 Compston, W. Interpretations of SHRIMP and isotope dilution zircon ages for the geological time-scale: 1. The early Ordovician and late Cambrian Mineralogical Magazine (2000) 64 (1): 43–57.

Reiners, P., Carlson, R., Renne, P., Cooper, K., Granger, D., McLean, N., Schoene, B. (2017). Interpretational approaches: making sense of data. In Geochronology and Thermochronology. Wiley. 10.1002/9781118455876.ch4.

Sambridge, M.S., Compston W.: Mixture modelling of multi-component data sets with application to ion-probe zircon ages. EPSL 128 373-390. 1994

Scientific Technical Report STR - Data 21/06 ISSN 2190-7110G-Chron 2019 – Round 1

Webb, P., Wiedenbeck, M., Glodny, J. An International Proficiency Test for U-Pb Geochronology Laboratories -Report on the 2019 Round of G-Chron based on Palaeozoic Zircon Rak-17 Submitted